# Choroidal and retinal thinning in chronic kidney disease independently associate with eGFR decline and are modifiable with treatment

Tariq E. Farrah[1,2], Dan Pugh[1,2], Fiona A. Chapman[1,2], Emily Godden ●[1,2], Craig Balmforth[1], Gabriel C. Oniscu[3,4], David J. Webb ●[1], Baljean Dhillon ●[5,6], James W. Dear ●[1], Matthew A. Bailey ●[1], Peter J. Gallacher[1] & Neeraj Dhaun ●[1,2,3] ✉

In patients with chronic kidney disease (CKD), there is an unmet need for novel biomarkers that reliably track kidney injury, demonstrate treatment-response, and predict outcomes. Here, we investigate the potential of retinal optical coherence tomography (OCT) to achieve these ends in a series of prospective studies of patients with pre-dialysis CKD (including those with a kidney transplant), patients with kidney failure undergoing kidney transplantation, living kidney donors, and healthy volunteers. Compared to health, we observe similar retinal thinning and reduced macular volume in patients with CKD and in those with a kidney transplant. However, the choroidal thinning observed in CKD is not seen in patients with a kidney transplant whose choroids resemble those of healthy volunteers. In CKD, the degree of choroidal thinning relates to falling eGFR and extent of kidney scarring. Following kidney transplantation, choroidal thickness increases rapidly (~10%) and is maintained over 1-year, whereas gradual choroidal thinning is seen during the 12 months following kidney donation. In patients with CKD, retinal and choroidal thickness independently associate with eGFR decline over 2 years. These observations highlight the potential for retinal OCT to act as a non-invasive monitoring and prognostic biomarker of kidney injury.

Chronic kidney disease (CKD) is a global health problem the prevalence of which increased by 30% worldwide between 1990 and 2017[1]. CKD is strongly associated with incident cardiovascular disease. Indeed, patients with pre-dialysis CKD are more likely to die from cardiovascular disease than develop kidney failure[1]. Kidney transplantation is the only treatment that reduces cardiovascular risk and prolongs the survival of patients with kidney failure but these patients remain at a 3–5-fold greater cardiovascular risk than the matched general population and cardiovascular disease is their leading cause of death[2].

A reduced estimated glomerular filtration rate (eGFR) is strongly associated with adverse cardiovascular and renal outcomes[3], and as

[1]Edinburgh Kidney, University/BHF Centre for Cardiovascular Science, The Queen's Medical Research Institute, University of Edinburgh, Edinburgh, UK. [2]Department of Renal Medicine, Royal Infirmary of Edinburgh, Edinburgh, UK. [3]Edinburgh Transplant Centre, Royal Infirmary of Edinburgh, Edinburgh, UK. [4]Transplant Division, Karolinska Institutet Stockholm, Stockholm, Sweden. [5]Centre for Clinical Brain Sciences, University of Edinburgh, Edinburgh, UK. [6]Princess Alexandra Eye Pavilion, Edinburgh, UK. ✉e-mail: bean.dhaun@ed.ac.uk

eGFR declines these risks increase[4]. However, eGFR lacks sensitivity and specificity. For example, substantial renal tissue damage has to occur before eGFR is impaired[5]. It is also recognized that the focus on eGFR-related outcomes is unidimensional, poorly reflecting complex disease pathology[6]. Microvascular changes are important in CKD development and progression. Currently, these can only be assessed reliably through kidney biopsy, which is invasive. Interval kidney biopsy, to assess microvascular changes over time and in response to therapies, is also impractical. Thus, there is an urgent unmet need for novel biomarkers that will sensitively and specifically track kidney injury, reliably demonstrate response to treatments, and predict longer-term outcomes.

The kidney and eye are structurally and functionally similar[7,8] meaning that diseases may present similarly and via common pathways in both organs. For example, Bruch's membrane in the eye and the glomerular basement membrane (GBM) both contain a network of α3, α4 and α5 type IV collagen chains[9,10]. Thus, inherited diseases of type IV collagen manifest with co-existent nephropathy and retinopathy as seen in Alport syndrome[11]. The microcirculation of the retina can be subdivided into retinal and choroidal circulations. The choroidal capillary (choriocapillaris) endothelium has ~80 nm fenestrations allowing fluid exchange within the subretinal space[12] similar to the glomerular endothelium for ultrafiltration into the urinary space[13]. In addition, the eye and kidney have matched chorioretinal and corticomedullary oxygen gradients, respectively, and excessive activation of the renin-angiotensin-aldosterone and endothelin systems are implicated in the development and progression of both retinopathy and CKD[14–17]. The choroidal circulation receives ~80% of ocular blood flow and passively oxygenates key visual apparatus including the pigment epithelium and photoreceptors particularly within the avascular fovea, suggesting a critical role in maintaining global retinal health[18]. Choroidal vascular change may therefore predate the onset of overt retinopathy and, if detectable, might allow earlier identification of incipient disease.

The transparency of the ocular media offers a unique opportunity to directly visualize the microvasculature within the eye which may be affected in systemic diseases such as CKD[19]. Optical coherence tomography (OCT) is a non-invasive and rapid method to cross-sectionally image the eye that is available in most high street opticians[8,20]. Recent advances have enabled the identification of specific cell layers within the retina in high resolution, as well as deeper structures such as the choroid, an exclusively vascular bed, in a way only previously possible histologically. Recently, we demonstrated, for the first time, that patients with CKD have choroidal and retinal thinning compared to matched healthy volunteers[21]. Here, in a series of prospective studies, we show that OCT metrics reflect circulating and histological measures of kidney injury, are modified by treatments for kidney disease and can predict future decline of kidney function. OCT metrics, therefore, have the potential to act as a non-invasive monitoring and prognostic biomarkers of kidney injury.

## Results

We enrolled (*i*) patients with pre-dialysis CKD, (*ii*) patients with kidney failure undergoing kidney transplantation, (*iii*) living kidney donors, and (*iv*) healthy volunteers into a series of prospective cross-sectional and longitudinal studies as shown in Fig. 1.

### Study 1: OCT metrics in health, CKD & following kidney transplantation

Three groups of subjects were recruited to this study: 112 patients with CKD due a range of underlying etiologies, 92 patients with a functional kidney transplant and 86 healthy volunteers. The three groups were matched for age and sex, and the CKD and transplant groups were matched in terms of eGFR. Patients with CKD had a greater degree of systemic inflammation−reflected by a higher high-sensitivity

C-reactive protein (hsCRP)−and heavier proteinuria than kidney transplant recipients, and patients with a kidney transplant had received their transplant on average ~7 years prior to OCT imaging. (Table 1).

The retina was thinner in patients with CKD compared to healthy volunteers and this thinning was particularly apparent in the central retina (~5% thinner compared to health) (Fig. 2A). In keeping with a thinner retina, macular volume was reduced in CKD (health vs. CKD: $8.73 \pm 0.36$ mm$^3$ vs. $8.44 \pm 0.44$ mm$^3$, $p < 0.001$) (Fig. 2B). Recipients of a kidney transplant also demonstrated retinal thinning and reduced macular volume compared to healthy volunteers and these were similar in magnitude to those seen in patients with CKD. Retinal nerve fibre layer (RNFL) thickness did not differ between the three groups (Fig. 2C). In CKD patients, increasing age and lower eGFR were associated with a reduced macular volume, $r = -0.30$, $p < 0.001$ and $r = 0.25$, $p = 0.02$, respectively (Supplementary Fig. 1A). The association between eGFR and macular volume persisted after adjustment for age (Supplementary Table 1). We did not demonstrate any difference in retinal thickness, macular volume and RNFL thickness between men and women.

The choroid was thinner in patients with CKD at each of the three macular locations assessed (health vs. CKD, locations I, II and III: $234 \pm 80$ μm vs. $197 \pm 85$ μm, $319 \pm 93$ μm vs. $274 \pm 93$ μm, $292 \pm 83$ μm vs. $262 \pm 84$ μm, $p < 0.01$ for each). Conversely, patients with a kidney transplant had choroidal thicknesses similar to healthy levels, suggesting reversal of this thinning (Fig. 3A). In those with CKD, the degree of choroidal thinning related to increasing age ($r = -0.25$, $p = 0.008$) and lower eGFR such that those patients with a lower eGFR had a thinner choroid ($r = 0.30$, $p = 0.001$, Fig. 3B and Supplementary Fig. 1B). As with macular volume, the relationship between lower eGFR and choroidal thinning persisted after adjustment for age (Supplementary Table 2). Tacrolimus, a calcineurin inhibitor, is part of standard kidney transplant immunosuppression protocols; it has several dose-dependent functional and structural effects on the vasculature including endothelial dysfunction and allograft hyaline arteriolopathy[22]. We found that choroidal thinning was more marked in those kidney transplant recipients with greater tacrolimus exposure (location I: $r = -0.34$, $p = 0.002$; location II: $r = 0.30$, $p = 0.007$; location III: $r = 0.32$, p = 0.004, Supplementary Fig. 2). We observed no differences in choroidal thickness between men and women in any of the three subject groups.

### Study 2: OCT metrics & kidney histological injury

Next, we examined associations between OCT metrics and kidney histological injury in those patients who had undergone a kidney biopsy within 30 days of their OCT scan ($n = 50$, Supplementary Table 3). Choroidal thinning at all three locations was independently associated with more extensive kidney scarring (Supplementary Fig. 3), specifically the severity of interstitial fibrosis with tubular atrophy, after adjusting for age, eGFR, glomerulosclerosis and hypertension (Supplementary Table 4). Macular volume did not associate with histological indices of kidney injury.

### Study 3: Reversal of choroidal thinning with gain of GFR & choroidal thinning with loss of GFR

Given our findings that patients with CKD have choroidal thinning that worsens as eGFR declines and that patients with a kidney transplant have a healthy choroidal thickness, we next examined the timing of these choroidal changes. Patients with kidney failure receiving a kidney transplant show rapid improvements in eGFR. Conversely, kidney donors rapidly lose eGFR when they undergo a nephrectomy. We utilized these clinical scenarios to assess the impact of changing eGFR on OCT metrics.

Twenty-five patients with kidney failure undergoing kidney transplantation from a living donor were recruited to this study

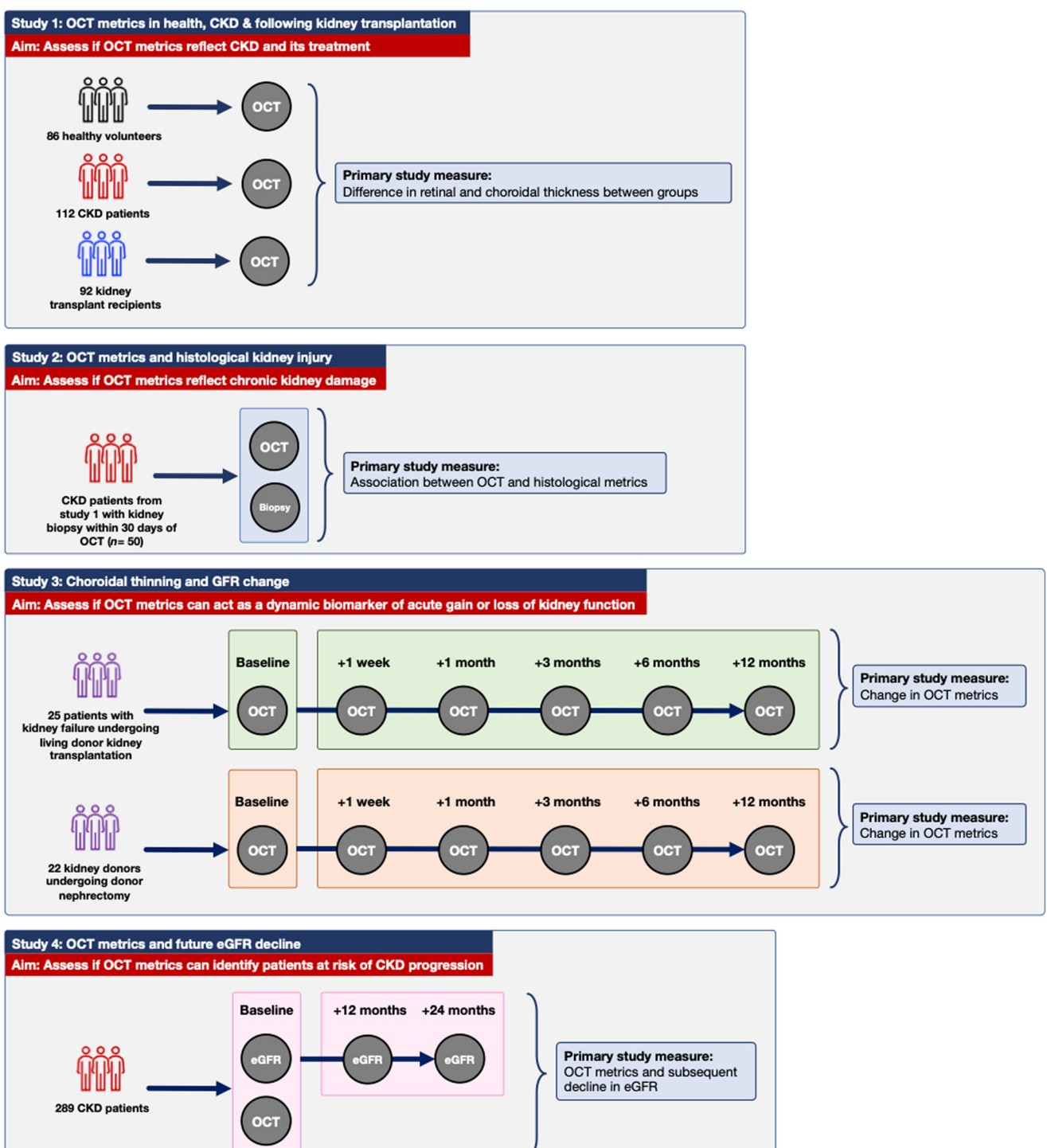

**Fig. 1 | Schematic overview of study participants, imaging schedule and primary outcomes.** OCT optical coherence tomography, CKD chronic kidney disease, eGFR estimated glomerular filtration rate.

(Supplementary Table 5). Overall, eGFR improved from $8 \pm 3$ ml/min/$1.73\,m^2$ to $58 \pm 21$ ml/min/$1.73\,m^2$ in the first week post-transplantation ($p < 0.001$), and this continued to improve over the 12-month study follow-up. We observed early thickening of the choroid in the post-transplant period with a ~5% increase by 1 week at each of locations I, II and III. At 1 month after transplant, choroidal thickness had increased by ~10% at each location (pre-transplant vs. 1-month post-transplant, location I: $230 \pm 79\,\mu m$ vs. $247 \pm 77\,\mu m$, $p = 0.03$; location II: $321 \pm 70\,\mu m$ vs. $359 \pm 78\,\mu m$, $p < 0.001$; location III: $290 \pm 74\,\mu m$ vs.

$318 \pm 65\,\mu m$, $p = 0.003$) (Fig. 4A). This increase in choroidal thickness was maintained up to 12 months after transplant (Supplementary Fig. 4). Kidney transplant recipients also had significant increases in both retinal thickness and macular volume up to 12 months following transplantation. However, there were no consistent changes in RNFL thickness (Fig. 4B and Supplementary Fig. 5).

Twenty-two kidney donors underwent OCT scanning prior to kidney donation and then at fixed intervals post-donation (Supplementary Table 5). Serum creatinine increased from $68 \pm 8\,\mu mol/L$ to

## Table 1 | Baseline characteristics for study 1

| | CKD | Transplant | Health |
|---|---|---|---|
| Number of participants (*n* = 290) | 112 | 92 | 86 |
| Age, years | 52 ± 13 | 50 ± 12 | 50 ± 13 |
| Male, *n* (%) | 67 (60) | 60 (65) | 47 (55) |
| Smoking status, *n* (%) | | | |
| Never | 64 (57) | 61 (66) | 67 (78) |
| Current | 14 (13) | 4 (4) | 5 (6) |
| Ex-smoker | 28 (25) | 27 (29) | 8 (9) |
| No data | 6 (5) | 0 (0) | 6 (7) |
| Primary renal diagnosis, *n* (%) | | | |
| ANCA vasculitis | 39 (35) | 1 (1) | – |
| Membranous | 15 (13) | 3 (3) | – |
| SLE | 7 (6) | 3 (3) | – |
| Polycystic kidney disease | 6 (5) | 17 (18) | – |
| IgA nephropathy | 5 (4) | 15 (16) | – |
| Hypertension | 5 (4) | 2 (2) | – |
| FSGS | 4 (4) | 2 (2) | – |
| Interstitial nephritis | 4 (4) | 7 (8) | – |
| Reflux nephropathy | 3 (3) | 7 (8) | – |
| Other | 7 (6) | 12 (13) | – |
| Unknown | 17 (15) | 17 (18) | – |
| Time from transplant, days | - | 2435 ± 2753 | – |
| Clinical | | | |
| BMI, kg/m² | 28.3 ± 4.7 | 25.9 ± 4.0 | 26.3 ± 3.3 |
| BP, mmHg | | | |
| Systolic | 131 ± 19 | 138 ± 17 | 128 ± 17 |
| Diastolic | 78 ± 11 | 82 ± 11 | 82 ± 12 |
| MAP | 95 ± 14 | 100 ± 11 | 97 ± 16 |
| Laboratory | | | |
| Creatinine, μmol/L | 145 ± 76 | 147 ± 81 | 72 ± 10 |
| eGFR, ml/min/1.73 m² | 55 ± 27 | 55 ± 24 | 97 ± 14 |
| GFR stage, ml/min/1.73 m² *n* (%) | | | |
| ≥90 | 15 (13) | 6 (7) | 61 (71) |
| 60–89 | 27 (24) | 30 (33) | 25 (29) |
| 30–59 | 46 (41) | 40 (43) | 0 (0) |
| 15–29 | 20 (18) | 14 (15) | 0 (0) |
| <15 | 4 (4) | 2 (2) | 0 (0) |
| No data | 0 (0) | 0 (0) | 0 (0) |
| Haemoglobin, g/L | 128 ± 18 | 130 ± 19 | 141 ± 10 |
| hsCRP, mg/L | 17 ± 24 | 6 ± 11 | 2 ± 5 |
| uPCR, mg/mmol | 201 ± 402 | 60 ± 74 | 5 ± 6 |
| Medications, *n* (%) | | | |
| Aspirin | 10 (9) | 17 (18) | 0 (0) |
| α-blocker | 10 (9) | 17 (18) | 2 (2) |
| ACE inhibitor | 39 (35) | 22 (24) | 2 (2) |
| Angiotensin receptor blocker | 19 (17) | 3 (3) | 0 (0) |
| β-blocker | 15 (13) | 30 (33) | 0 (0) |
| Calcium channel blocker | 29 (26) | 42 (46) | 1 (1) |
| Diuretic | 13 (12) | 4 (4) | 0 (0) |
| Statin | 51 (46) | 34 (37) | 1 (1) |

*ACE* angiotensin converting enzyme, *ANCA* anti-neutrophil cytoplasm antibody, *BMI* body mass index, *BP* blood pressure, *eGFR* estimated glomerular filtration rate, *FSGS* focal and segmental glomerulosclerosis, *hsCRP* high sensitivity C-reactive protein, *IgA* immunoglobulin A, *MAP* mean arterial pressure, *uPCR* urine protein:creatinine ratio, *SLE* systemic lupus erythematosus

105 ± 23 μmol/L and eGFR fell from 97 ± 10 ml/min/1.73 m² to 62 ± 14 ml/min/1.73 m² in the first week post-donation and these remained stable over the 12-month follow-up period. Following this acute loss of functional kidney mass, the choroid thickened in the first week post-kidney donation before showing a tendency to thinning over the longer term at each of locations I, II and III (pre-donation vs. 12 months post-donation, location I: 226 ± 66 μm vs. 212 ± 69 μm; location II: 320 ± 74 μm vs. 291 ± 80 μm; location III: 305 ± 60 μm vs. 284 ± 58 μm, Fig. 5A and Supplementary Fig. 6A). We observed no consistent changes following kidney donation in macular volume (Fig. 5B), retinal or RNFL thickness (Supplementary Fig. 6B).

### Study 4: Macular volume & choroidal thickness independently associate with eGFR decline in CKD

Finally, we determined whether macular volume (as a metric of retinal thickness) and choroidal thickness associated with eGFR decline in patients with stable CKD. OCT images were available for 289 subjects with CKD. Five (2%) patients were excluded as their OCT scans were of insufficient quality, 16 (6%) were excluded because of eye disease identified at the time of imaging, and 6 patients (2%) were excluded due to insufficient follow-up (Supplementary Fig. 7). Thus, we included 262 patients (91%; mean age 57 ± 14 years; 41% female) in the final analysis (Supplementary Table 6). Patients' baseline eGFR was 42 ml/min/1.73 m², proteinuria was -0.5 g/day, and BP was controlled to current guidelines (136/78 mmHg). Across all patients (*n* = 262), the mean macular volume was 8.3 ± 0.4 mm³ and mean choroidal thicknesses were 203 ± 81 μm, 276 ± 85 μm and 257 ± 77 μm at locations I, II and III, respectively.

The primary outcomes of an eGFR decline of ≥10% at 1 year and ≥20% at 2 years occurred in 38% (100/262) and 23% (60/262) of patients, respectively. The results of unadjusted analyses evaluating the relationship between each OCT metric and these outcomes are summarized in Supplementary Table 7. For every 1 mm³ decrease in macular volume, the odds of eGFR declining by ≥10% at 1 year increased by 2.48 (95% CI 1.26 to 5.08; *p* = 0.01), whilst the odds of eGFR declining by ≥20% at 2 years increased by 3.75 (95% CI 1.26 to 5.08; *p* = 0.004). Of the three choroidal locations assessed, only location III demonstrated a significant association with each primary outcome. Indeed, for every 10 μm decrease in choroidal thickness at location III, the odds of an eGFR decline of ≥10% at 1 year and ≥20% at 2 years increased by 1.04 (95% CI 1.01 to 1.08; *p* = 0.02) and 1.06 (95% CI 1.01 to 1.10; *p* = 0.01), respectively.

Multivariable logistic regression models – adjusting for variables that are recognized to impact upon eGFR decline specifically, patient age, sex, baseline eGFR, systolic BP and proteinuria—further evaluated the relationships between each OCT metric and the study outcomes on the basis that this combination of covariates in a logistic regression model provided the best fit for the data (Supplementary Tables 8, 9). Consistent with the unadjusted analyses, the fully adjusted models demonstrated a significant increase in the odds of an eGFR decline of ≥10% at 1 year for every 1 mm³ decrease in macular volume (OR 2.41, 95% CI: 1.07 to 5.70; *p* = 0.04), and for every 10 μm decrease in choroidal thickness at location III (OR 1.06, 1.01 to 1.11; *p* = 0.02) (Table 2). We additionally observed a large increase in the odds of an eGFR decline of ≥20% at 2 years for every 1 mm³ decrease in macular volume (OR 2.42, 0.94 to 6.57; *p* = 0.07) and more modest increases for every 10 μm decrease in choroidal thickness at locations I (OR 1.01, 0.96 to 1.05; *p* = 0.82) and III (OR 1.04, 0.98 to 1.09; *p* = 0.19) (Table 2).

## Discussion

Estimated glomerular filtration rate is a key indicator of kidney function in patients with CKD. However, it is insensitive, only becoming abnormal once >50% of functional kidney mass has been lost[5]. Whilst the degree of scarring on kidney biopsy (for example, interstitial fibrosis with tubular atrophy) is a more reliable metric of kidney disease severity and prognosis[23], its use in routine clinical practice is impractical. Retinal imaging offers an attractive route to non-invasively assess microvascular remodelling in chronic systemic disease. In these prospective

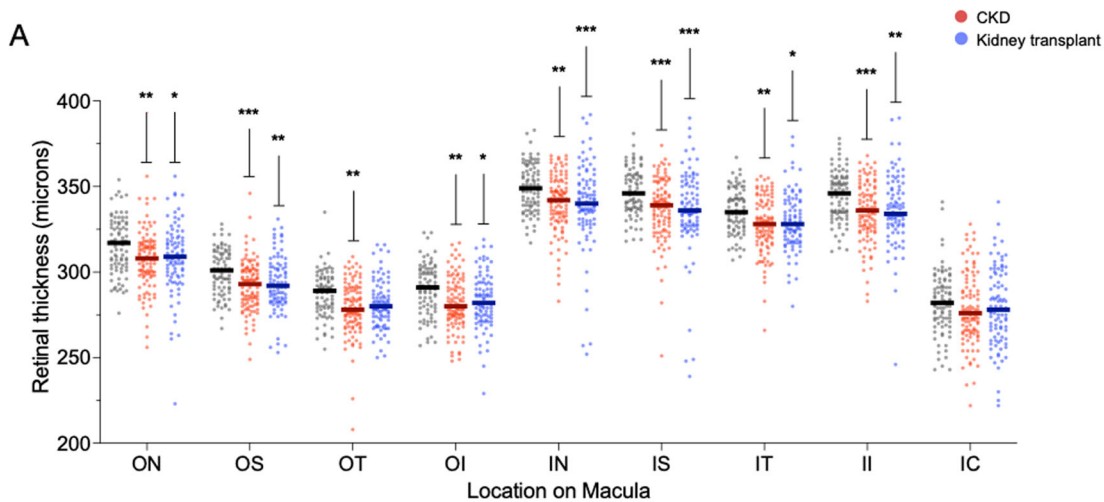

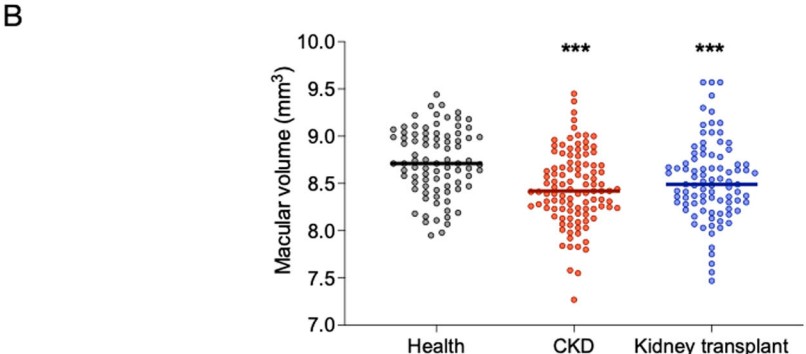

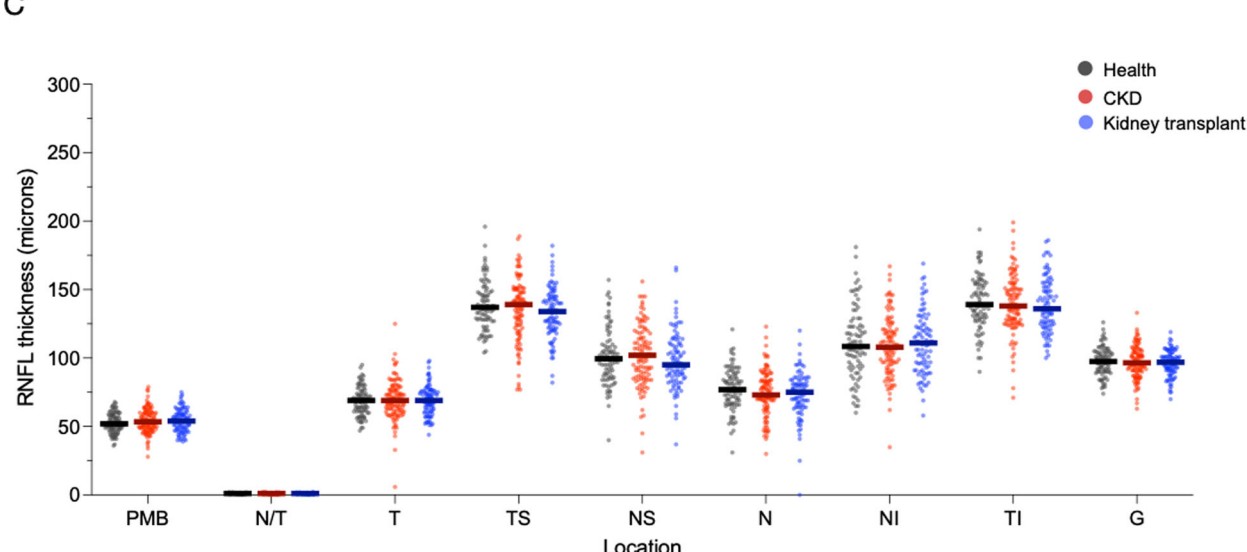

**Fig. 2 | Retinal and nerve fibre layer thickness in health and kidney disease.**
Scatter dot plots of retinal thickness (**A**), macular volume (**B**) and retinal nerve fibre layer thickness (RNFL, **C**) of healthy volunteers (grey, $n = 86$), patients with chronic kidney disease (CKD, red, $n = 112$) and patients with a kidney transplant (blue, $n = 92$) at different areas across the macula as defined in Supplementary Fig. 1. For retinal thickness (**A**): ON outer nasal *$p = 0.02$, **$p = 0.009$; OS outer superior **$p = 0.002$, ***$p < 0.001$; OT outer temporal **$p = 0.004$; OI outer inferior *$p = 0.04$, **$p = 0.003$; IN inner nasal **$p = 0.002$, ***$p < 0.001$; IS, inner superior ***$p < 0.001$; IT inner temporal *$p = 0.04$, **$p = 0.004$; II inner inferior **$p = 0.006$, ***$p < 0.001$; IC inner circle. For macular volume, ***$p < 0.001$. For RNFL (**C**) T temporal, TS temporal-superior, NS nasal-superior, N nasal, NI nasal-inferior, TI temporal-inferior. PMB papillo-macular bundle, N/T nasal-temporal ratio, G average RNFL thickness. Lines represent mean. $p$ values are vs. healthy volunteers. Two-sided analysis by ANOVA with Tukey correction for multiple comparisons. Source data are provided as a source data file.

studies, we assessed the potential for retinal OCT metrics to report kidney function during CKD progression, and in response to treatment. Our major findings are that: (1) retinal and choroidal thinning occurs in CKD and progresses as kidney function declines; (2) these changes in OCT metrics are reversed when kidney function is restored by kidney transplantation; (3) healthy individuals who donate a kidney and lose kidney function, gradually develop choroidal thinning and, (4) in those with CKD, a thinner retina and choroid seen on a single point-in-time OCT scan independently associate with future eGFR decline. These observations highlight the potential for OCT metrics to act as a non-invasive monitoring and prognostic biomarker of kidney injury.

The greater the severity of kidney disease (reflected by a lower eGFR), the thinner the retina and choroid. Notably, this was independent of age, a recognized, important contributor to chorioretinal thinning[24]. We also found that the retinal thinning in patients with CKD was more marked in the central retina, compared with the peripheral scan regions. The central retina is reliant on the choroidal circulation for oxygen supply[8], which may make it more vulnerable to microvascular injury. Indeed, choroidal thickness was reduced by ~15% in CKD across the three locations examined and related directly to kidney scarring even after adjusting for important confounders. This relationship was stronger than that between choroidal thickness

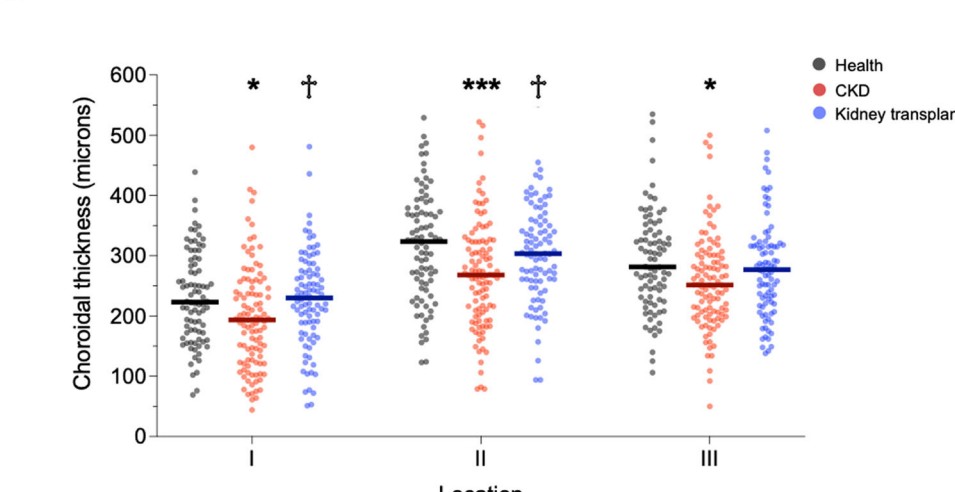

**Fig. 3 | Choroidal thickness in health and kidney disease. A** Scatter dot plots of choroidal thickness in healthy volunteers (grey, $n = 86$), patients with chronic kidney disease (CKD, red, $n = 112$) and patients with a kidney transplant (blue, $n = 92$) at location I (2 mm nasal to fovea), location II (subfoveal) and location III (2 mm temporal to fovea). Lines represent mean. Patients with CKD vs. health volunteers: location I *$p = 0.014$, location II ***$p < 0.001$, location III *$p = 0.021$; patients with a kidney transplant vs. patients with CKD, location I †$p = 0.031$, location II †$p = 0.012$.

Two sided analysis by ANOVA with Tukey correction for multiple comparisons. **B** Scatter dot plots of subfoveal choroidal thickness (upper panel) grouped by KDIGO CKD stage as defined by estimated glomerular filtration rate (eGFR, lower panel). Red−patients with CKD stage 4/5, $n = 26$; Orange−patients with CKD stage 3, $n = 47$; Yellow−patients with CKD stage 2, $n = 28$; Green−patients with CKD stage 1, $n = 14$. Lines represent mean. Two-sided analysis by ANOVA for linear trend, $p = 0.004$. Source data are provided as a source data file.

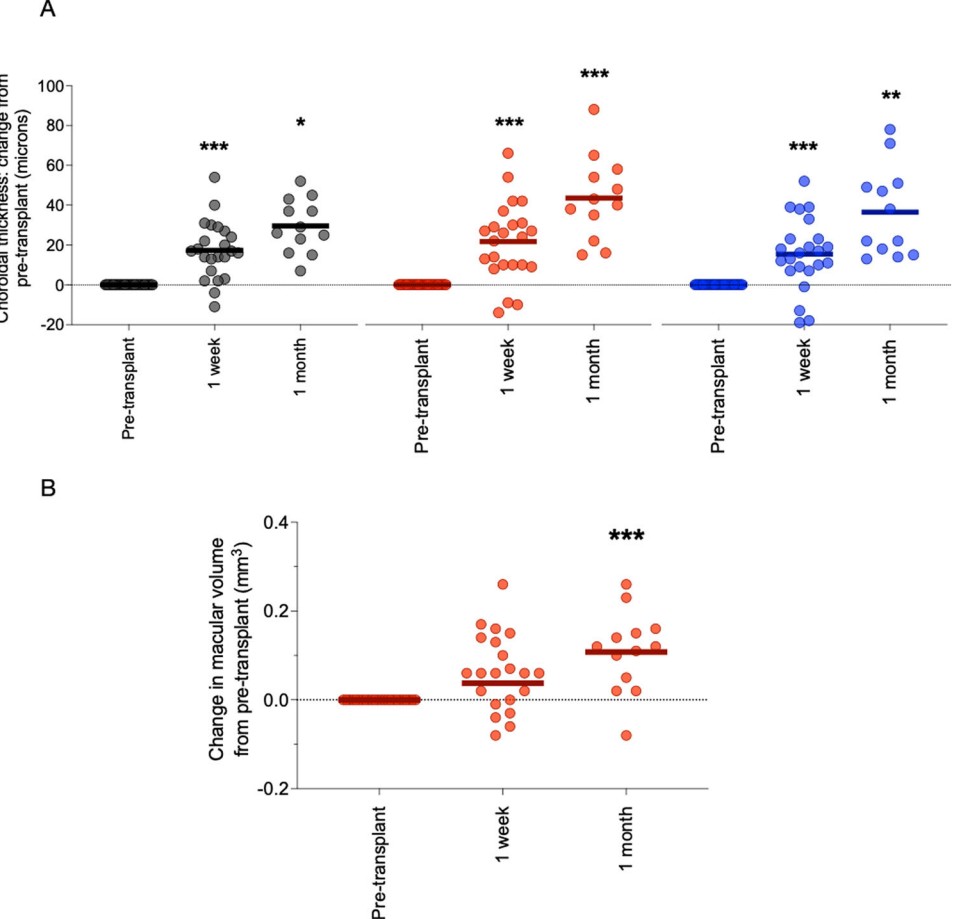

**Fig. 4 | OCT metrics and kidney transplantation.** Dot plots of change in pretransplant choroidal thickness (**A**) and macular volume (**B**) at 1 week and 1 month after living donor kidney transplantation. Grey—choroidal thickness 2 mm nasal to fovea; red – subfoveal choroidal thickness; blue—choroidal thickness 2 mm temporal to fovea. Lines represent mean. For choroidal thickness, at 1 week, ***$p < 0.001$ vs. pre-transplant; at 1 month, *$p = 0.030$, **$p = 0.003$, ***$p < 0.001$ vs. pretransplant. $n = 25$. Analysis by mixed effects model with Sidak correction for multiple comparisons. For macular volume, ***$p < 0.001$. Analysis by mixed-effects model with Sidak correction for multiple comparisons. Source data are provided as a source data file.

and eGFR, suggesting the potential for non-invasive OCT to convey important clinical information about patients with kidney disease. We found that choroidal thinning was particularly evident in the subfoveal region (location II), which might account for the exaggerated retinal thinning observed centrally around the fovea. From a biomarker perspective, a focus on these central areas may further improve sensitivity of OCT metrics to report microvascular injury in CKD.

One of the main aims of our study was to examine if the OCT changes seen in CKD are dynamic and modifiable. Two complementary paradigms show these to be the case. First, we show that patients with kidney failure receiving a kidney transplant undergo rapid thickening of the retina and choroid, an effect that begins as early as 1 week following transplant and continues for at least 12 months. Our data suggest a generalised increase in retinal thickness and thus macular volume, rather than focal accumulation of intra or sub-retinal fluid, which appears more evident in central regions. The -2% increase in macular volume represents a deviation *toward* values observed in health and so may not portend the same pathological significance as increases observed in age-related macular degeneration or diabetic macular oedema that are deviations *away* from health driven by underlying vasculopathy. The implications of these increases for the ocular health of kidney transplant recipients are beyond the scope of our study but will be an important part of future work as ocular disease is common in these patients[25].

The most impressive increase in thickness was observed in the choroid, suggesting that the changes here are reporting improving vascular flow and/or function. This is an important finding; kidney transplantation is the only treatment that reduces cardiovascular risk in patients with kidney failure[2]. Nevertheless, cardiovascular disease remains the leading cause of death in transplant recipients[26]. Conventional cardiovascular risk factors, such as hypercholesterolemia and proteinuria, are poor predictors of cardiovascular events in this setting[27] and OCT imaging may offer a route to robustly assess cardiovascular risk in this high-risk population. Further support of a link between choroidal thickness and vascular function is provided by our data showing a strong association between tacrolimus exposure and choroidal thickness. Tacrolimus, a calcineurin inhibitor and a standard component of post-transplant immunosuppression, is well recognized to promote vasoconstriction, endothelial dysfunction, and arterial hypertension; those patients with a greater tacrolimus burden had thinner choroids[28].

The second key observation came from serial OCT scans in living kidney donors, a group in whom GFR is rapidly lost. Living kidney donors demonstrated choroidal thinning over a 12-month period even though eGFR remained within the normal reference range during this follow-up period. Kidney donation is associated with important longer-term risks of developing both chronic kidney and cardiovascular disease[29,30]. Currently, we are unable to identify kidney donors most at risk of these complications. Recently, we showed

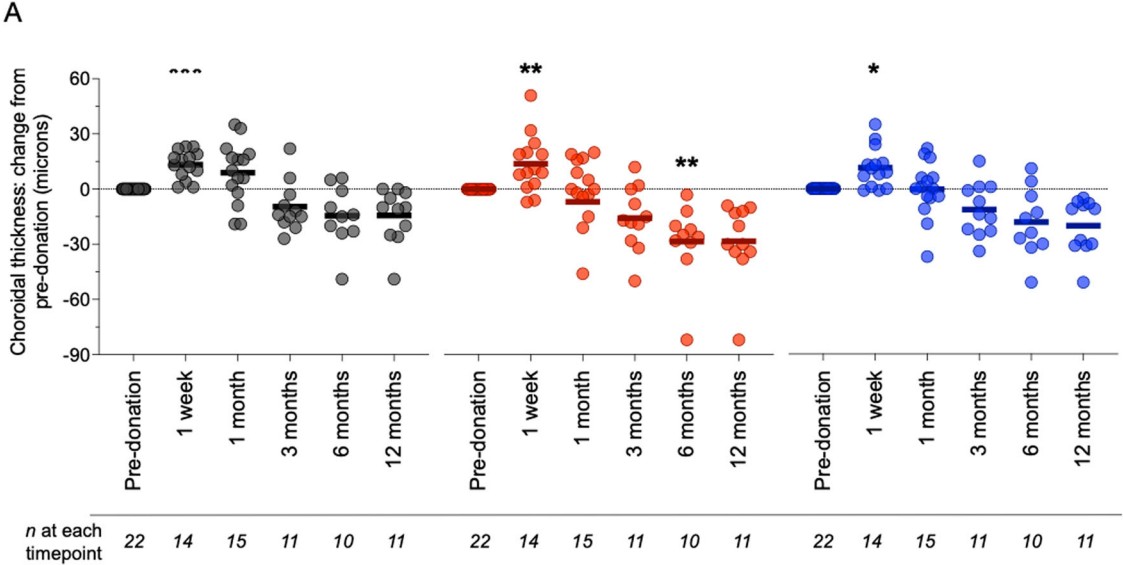

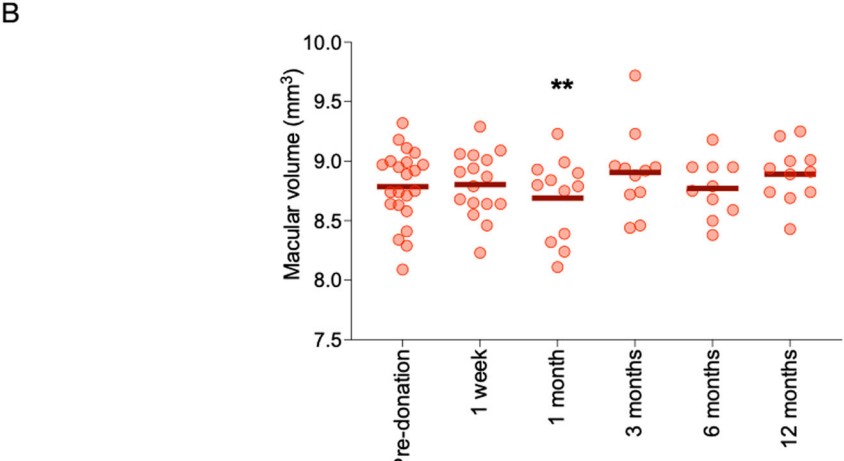

**Fig. 5 | OCT metrics and kidney donation.** Dot plots of change in choroidal thickness (**A**) and macular volume (**B**) at 1 week, 1, 3, 6 and 12 months after donor nephrectomy. Grey−choroidal thickness 2 mm nasal to fovea; red−subfoveal choroidal thickness; blue−choroidal thickness 2 mm temporal to fovea. Lines represent mean. For choroidal thickness at 1 week, *p = 0.01, **p = 0.002, *** $p < 0.001$ vs. pre-donation. For choroidal thickness at 6 months, **p = 0.004 vs. pre-donation. n = 22. Analysis by mixed-effects model with Sidak correction for multiple comparisons. For macular volume, **p = 0.009 vs. pre-donation. Analysis by mixed-effects model with Sidak correction for multiple comparisons. Source data are provided as a source data file.

the potential utility of OCT-angiography (OCT-A)−which defines retinal vascular density and geometry−in discriminating kidney donors with normal kidney function from matched healthy volunteers[31]. The phenotype of the donors' retinal vasculature evolved over time following donation to resemble that of patients with CKD. The current data suggest that OCT, which provides complementary data to OCT-A, may also have a role in long-term risk stratification of kidney donors, especially if future studies link OCT metrics to cardiovascular disease risk.

Importantly, our data suggest that factors other than eGFR influence OCT metrics in the longer-term. In our cross-sectional study, patients with CKD were matched to transplant recipients in terms of age, sex and eGFR (both ~55 ml/min/1.73 m²) but those with CKD had thinner choroids than transplant recipients. Patients with a kidney transplant had received their transplant on average ~7 years prior to OCT imaging. Notably, CKD patients had greater degrees of systemic inflammation and proteinuria than kidney transplant recipients at the time of imaging, and their underlying causes for developing CKD were more 'inflammatory' than those of transplant patients (e.g., anti-

neutrophil cytoplasm associated vasculitis, systemic lupus erythematosus). We have previously shown that both inflammation and proteinuria associate with choroidal thinning[21], and these two important contributors to endothelial dysfunction and cardiovascular risk[32] may partly explain the findings here. Indeed, the relationship between changing choroidal thickness and potential modification of cardiovascular risk remains to be defined and future prospective studies should aim to modulate inflammation and proteinuria in patients with CKD and examine the effects on OCT metrics.

We found no relationship between kidney disease (at any level of eGFR) and retinal nerve fibre layer (RNFL) thickness, and this is in line with our previous work[21]. The RNFL is the innermost layer of the retina and is made up of unmyelinated axons that form the optic nerve. RNFL thinning can therefore indicate axonal or neuronal atrophy in optic neuropathies or neurodegenerative conditions. Optic neuropathy is a rare complication of kidney disease[33]. Nevertheless, previous studies using OCT have shown RNFL thinning in patients with kidney failure receiving regular dialysis[34] and in those with earlier stages of CKD[33]. A complicating factor is the co-existence of diabetes mellitus in these

**Table 2 | Summary of multivariable logistic regression models evaluating relationship between each chorioretinal metric (total macular volume and choroidal thickness [locations I, II and III]) and the primary outcomes of a decline in eGFR of ≥10 at 1 year and ≥20% at 2 years**

| | Standard error | z-value | OR | Lower 95% CI | Upper 95% CI | p | Residual deviance |
|---|---|---|---|---|---|---|---|
| **Decline in eGFR of ≥10% at 1 year** | | | | | | | |
| Total macular volume (per 1 mm³ decrease) | 0.424 | −2.08 | 2.42 | 1.07 | 5.70 | 0.041 | 240 on 195 df |
| Choroid (location I) (per 10 μm decrease) | 0.002 | −0.30 | 1.01 | 0.97 | 1.05 | 0.763 | 244 on 195 df |
| Choroid (location II) (per 10 μm decrease) | 0.002 | −0.99 | 1.02 | 0.98 | 1.06 | 0.326 | 243 on 195 df |
| Choroid (location III) (per 10 μm decrease) | 0.002 | −2.38 | 1.06 | 1.01 | 1.11 | 0.026 | 238 on 195 df |
| **Decline in eGFR of ≥20% at 2 years** | | | | | | | |
| Total macular volume (per 1 mm³ decrease) | 0.492 | −1.79 | 2.42 | 0.94 | 6.57 | 0.071 | 175 on 150 df |
| Choroid (location I) (per 10 μm decrease) | 0.002 | −0.22 | 1.01 | 0.96 | 1.05 | 0.821 | 177 on 149 df |
| Choroid (location II) (per 10 μm decrease) | 0.002 | 0.38 | 0.99 | 0.95 | 1.04 | 0.705 | 177 on 149 df |
| Choroid (location III) (per 10 μm decrease) | 0.003 | −1.32 | 1.04 | 0.98 | 1.09 | 0.198 | 175 on 149 df |

Two sided analyses. Models were adjusted for patient age, sex, and baseline renal function, systolic blood pressure and urinary PCR.
*CI* confidence interval, *df* degrees of freedom, *eGFR* estimated glomerular filtration rate, *OR* odds ratio.

studies, which is a recognized cause of RNFL changes[35]. Our study purposefully focussed on non-diabetic CKD and, in the absence of other co-morbidity, it is likely that optic neuropathy would take time to develop in these patients. Additionally, we found no change in RNFL thickness in patients with kidney failure after they had received a kidney transplant or in those with a long-standing transplant, contrasting with a previous study which found RNFL thinning in 75% of kidney transplant recipients[25]. Again, differences in patient demographics and inclusion criteria might explain this.

Finally, we have shown that OCT metrics associate with eGFR decline in those with CKD. Currently, baseline eGFR, BP and proteinuria are the only reliable predictors of kidney function decline in CKD on a population scale. Using these parameters to individualize risk is not part of current standard of care. Here, we studied a cohort of patients that had modestly impaired kidney function at baseline (eGFR 42 ml/min/1.73 m²), with both BP and proteinuria controlled to current guidelines (-0.5 g/day and 137/78 mmHg, respectively). Thus, this was a cohort at low risk of disease progression. Despite this, our fully adjusted models show that for every 1 mm³ fall in macular volume (used as a metric of retinal thickness), the likelihood of an eGFR decline of ≥10% at 1 year increased 2.5-fold. Choroidal thickness was also independently associated with eGFR decline, although the effect was more modest (-10% increased risk of a ≥10% decline in eGFR at 1 year). We speculate that these effects may be even more impressive in patients at higher risk of disease progression. Notably, reduced macular volume showed a stronger association with risk of eGFR decline that then observed with choroidal thinning. The vascular nature of the choroid means that choroidal thickness is potentially subject to influence by blood pressure, sympathetic nervous system activation, volume status and vasoactive medications. These factors serve to increase the heterogeneity of choroidal thickness in any given population, and it would be impractical to consider and adjust for of all these in our analyses given our study size. Despite this, we still found an important association between choroidal thickness and eGFR decline which strengthens the utility of choroidal thickness as a biomarker for adverse kidney disease outcomes.

Patients with kidney disease are commonly prescribed medications such as angiotensin-converting enzyme inhibitors, β-blockers, and statins and these may affect the OCT parameters studied. However, in our cross-sectional study, all patients had been stabilized on their therapies and this is an unavoidable limitation of such studies. In our longitudinal study, patients undergoing kidney transplantation were started on several new medications following transplant including antimicrobials and immunosuppressants. However, we

studied the same patients over time, and for immunosuppressants, we examined the impact of tacrolimus exposure on our OCT metrics of interest.

We excluded patients with diabetes mellitus, a leading cause of CKD worldwide and so our findings are not generalisable to this highly prevalent patient group. This was because concomitant diabetic eye disease (retinopathy, maculopathy and choroidopathy) would confound interpretation of changes related to kidney disease per se. Indeed, choroidal thickening has been reported in diabetics even in the absence of overt retinopathy[36]. Additionally, treatments for diabetic eye disease such as pan-retinal laser photocoagulation[37] and anti-vascular endothelial growth factor therapy[38] directly affect retinal and choroidal thickness thus would have introduced further confounding. The majority of patients studied were of white northern European ethnicity reflecting our local population and thus our results may not be generalisable to other ethnic groups. There are some methodology points of note. The undulating nature of the choroidal-scleral interface limits the power of single-point choroidal thickness measurement and so we measured choroidal thickness at three locations. However, future studies might focus on choroidal volume which might better quantify overall disease burden.

Kidney disease is common, progressive, and inextricably associated with cardiovascular disease. 60,000 people die prematurely in the UK every year because of CKD and its associated cardiovascular disease. CKD costs the NHS - £2 billion per year. Half this cost is for provision of dialysis for -2% of patients. Currently, 64,000 patients are on dialysis in the UK[39] and the need for renal transplants far outstrips organ supply.

Incorporating metrics from the analysis of retinal photographs improves the accuracy of risk stratifying patients with CKD for disease progression and incident cardiovascular risk by -10%[40]. OCT metrics may plausibly be even more useful given the high-fidelity, granular nature of the images acquired, particularly if coupled with modern deep learning image analysis. OCT metrics have already been incorporated into clinical trial outcome measures in multiple sclerosis[41] and our data introduce the exciting possibility that these metrics might be included alongside eGFR and proteinuria to individualize cardiovascular risk and the risk of kidney disease progression in future trials in CKD.

## Methods
Between September 2016 and August 2020, we enrolled (1) patients with pre-dialysis CKD, (2) patients with kidney failure undergoing kidney transplantation, (3) living kidney donors, and (4) healthy

volunteers into a series of prospective cross-sectional and longitudinal studies (Fig. 1). All studies were carried out at the University of Edinburgh according to the principles of the Declaration of Helsinki. They were approved by the South-East Scotland ethics committee and were performed with written informed consent from each subject. The OCT And NEphropathy (OCTANE) study is registered at ClinicalTrials.gov: NCT02132741.

*Study 1* was a cross-sectional study that assessed differences in OCT metrics between subject groups. Specifically, those with CKD, those with a functional kidney transplant and healthy volunteers. For this study, subjects underwent a single OCT examination of both eyes. *Study 2*, a cross-sectional study, examined associations between OCT metrics and histological indices of kidney damage in a subgroup of CKD patients from *Study 1* who had a percutaneous native kidney biopsy within 30 days of OCT imaging. *Study 3* compromised two parallel longitudinal studies and evaluated how OCT metrics changed over time in patients with kidney failure undergoing kidney transplantation and, in those donating a kidney. For these studies, subjects underwent OCT examinations at baseline (the day before surgery) and then 1 week, 1-, 3-, 6- and 12-months following surgery. *Study 4* was a prospective cohort study and examined whether OCT metrics associate with future decline in kidney function as determined by estimated glomerular filtration rate (eGFR) (Fig. 1).

We additionally performed two sub-studies of healthy volunteers to assess the effects of different operators and time of day on variability of OCT metrics (Supplementary Fig. 8). *OCT scanning: intra- & inter-operator variability:* Fifteen healthy subjects were recruited to this study. Using our established study protocol, subjects underwent OCT scanning on two separate occasions at least 1 week apart by two different trained operators. For both intra-and inter-observer variability, OCT metrics from the first scan were compared with those of the second scan and across operators. We found no significant intra- or inter-operator variability. *OCT scanning: time of day* Twenty healthy subjects were recruited to this study and underwent OCT scanning at 0900, 1200 and 1600 on the same day and by the same operator. We found no significant differences in OCT metrics at any of the three timepoints.

All studies were performed in a quiet, temperature-controlled room. Systolic and diastolic blood pressure (BP) were recorded in duplicate, with an appropriately sized cuff, using a validated oscillometric sphygmomanometer, the Omron HEM-705CP, and values are presented as the average of two recordings[42]. Height and weight were recorded followed by OCT scanning. Blood and urine samples were taken at the end and stored at −80 °C. Venous blood samples were collected from all participants in EDTA and serum tubes. Samples were centrifuged immediately (EDTA tubes 2000 g for 15 min at 4 °C; serum tubes 2500 g for 10 min at 4 °C). Creatinine clearance, as an estimate of GFR, was calculated according to the Chronic Kidney Disease Epidemiology Collaboration (CKD-EPI) equation[43]. Tacrolimus exposure was estimated from the area under the curve of consecutive trough plasma tacrolimus concentrations against time in days since starting tacrolimus up to the date of the OCT study for each recipient of a kidney transplant and expressed as ng/L*days. Tacrolimus is a calcineurin inhibitor used as part of post-transplant immunosuppression.

All subjects underwent examinations of both eyes using the Heidelberg SPECTRALIS Spectral-Domain OCT machine (software version 6.16.2, Heidelberg Engineering). Images of the right eye were used for analysis wherever possible. Our protocol included three scans for each eye: (1) a horizontal line scan through the macula, centred over the fovea, with enhanced depth imaging (EDI) enabled for greater choroidal visualization; (2) a macular volume scan consisting of 61 horizontal B-scans with a separation of 120 μm covering the whole area of the macula; (3) a peripapillary circular line scan centred over the optic disc, with Nsite Axonal Analytics software

automated segmentation of the RNFL[44]. Images were acquired using proprietary TruTrack active eye tracking and Automatic Real-Time (ART) software, which averages the image over 100 scans, to generate a single high-resolution image.

Using OCT, we measured retinal thickness, macular volume (a summary metric of retinal thickness defined as the product of retinal thickness and scan area), choroidal thickness and RNFL thickness (Supplementary Fig. 9). Assessment of retinal thickness was performed according to the Early Treatment Diabetic Retinopathy Study (ETDRS) protocol[45]. The ETDRS map divides the macula into 9 subfields. The circular grid is centred over the fovea and consists of three concentric rings of diameters 1, 3, and 6 mm, respectively. The inner and outer rings are further divided into quadrants: temporal, nasal, superior, and inferior. Retinal thickness, RNFL thickness, and macular volume were measured using the automatic segmentation values of the SPECTRALIS OCT. Choroidal thickness was measured manually by two independent, blinded operators, on the horizontal EDI line scan, in three separate locations: 2 mm nasal to the fovea (location I), subfoveal (location II), and 2 mm temporal (location III) to the fovea. The measurement was taken in a vertical line from the outer hyper-reflective line corresponding to the base of the retinal pigment epithelium (RPE/basement membrane complex), to the choroidal-scleral junction (Supplementary Fig. 2).

Of the CKD subjects recruited, 50 had undergone kidney biopsy within 30 days of their OCT scan (Supplementary Table 3). All biopsies were suitable for histological assessment and were independently reviewed by two investigators blinded to OCT metrics and clinical details. Fibrosis was assessed on light microscopy on H&E-stained sections. The proportion of glomerulosclerosis and interstitial fibrosis and tubular atrophy (IFTA) were assessed across the whole biopsy specimen.

*CKD:* patients with stable pre-dialysis CKD, defined according to the Kidney Disease Outcome Quality Initiative (K/DOQI) classification[46], were recruited from the general renal outpatient clinics at the Royal Infirmary of Edinburgh. Patients were well and had not been started on any new medications in the 4 weeks leading up to the study. *Kidney transplant recipients & donors:* patients with a functional kidney transplant and living kidney donors were recruited from the transplant inpatient and outpatient services. For recipients, both those who had received a living and deceased donor (donation after both brain and circulatory death (DBD and DCD)) kidney transplant were included into the study. *Healthy volunteers:* were recruited from local, ethically approved databases. Subjects with any eye disease, previous eye surgery, refractive error greater than ±6 dioptres and those with diabetes mellitus were excluded from the study.

## Study 1: OCT metrics in health, CKD and kidney transplantation
The primary endpoint of this study was difference in choroidal thickness between CKD patients and healthy controls. Secondary endpoints included difference in retinal and RNFL thickness. This study was powered on the basis of a healthy subfoveal choroidal thickness of $289 \pm 52$ μm and a healthy temporal RNFL thickness of $79 \pm 16$ μm[47,48]. To detect a difference of 10% in choroidal thickness between healthy controls and patients with CKD, or in those with a functional kidney transplant, with 80% power and a two-sided significance of 5%, we calculated that we would need to recruit 82 participants in each group. To detect a similar magnitude of change in RNFL thickness, we would require 63 subjects in each group.

## Study 2: OCT metrics and histological kidney injury
The primary endpoint of this study was association between choroidal thickness and kidney scarring reflected by extent of glomerulosclerosis and interstitial fibrosis and tubular atrophy. This study was powered based on our previous work, with 40 patients providing 80 percent probability to detect a relationship between kidney scarring

and choroidal thickness at two-sided significance of 5%, assuming a standard deviation of choroidal thickness of ±50 um.

### Study 3: OCT metrics and change in acute GFR change

The primary endpoint was a difference in OCT metrics 12 months after surgery compared to pre-surgery. The studies were designed to detect differences in OCT metrics with 80% power and with a two-sided significance of 5%. *Kidney transplant recipients:* our previous work in patients with kidney failure showed a choroidal thickness of $241 \pm 50\,\mu m$ and an RNFL thickness of $48 \pm 11\,\mu m^2$[21]. To detect a 15% change in choroidal thickness in patients with kidney failure undergoing kidney transplantation, we would need 24 subjects. To detect the same change in RNFL thickness, 19 subjects would be required. *Kidney donors:* to detect a 15% change in choroidal thickness in a healthy subject pre- and post-kidney donation, we needed to recruit 21 subjects. To detect a similar magnitude of change in RNFL thickness, 20 subjects were needed.

### Study 4: Association of OCT metrics with eGFR decline in patients with pre-dialysis CKD

For this study, our primary outcomes were a decline in eGFR of ≥10% at 1 year and ≥20% at 2 years. Linked, routinely collected, individual-level biochemistry data were accessed from our regional registry of renal patients, *VitalData*. The 'index' or baseline eGFR was defined as the first test result available within 6 months of the OCT scan. One-year outcomes were calculated based on the first available measure of eGFR during a 6-month window 1 year following the index eGFR. Two-year outcomes were calculated based on the first available measure of eGFR during a 6-month window 1 year following the result used to define the 1-year outcome. The study endpoints were selected based on the results of a meta-analysis of >1.7 million patients, which demonstrated that reductions in eGFR such as these were frequently and robustly associated with subsequent risk of kidney failure in patients with and without renal impairment at baseline[49]. From our previous work[21,50], we estimated that the standard deviation of baseline eGFR in our population would be ~25 mL/min/1.73 m$^2$ and that the difference in mean eGFR at 1 year would be ~5 mL/min/1.73 m$^2$. Therefore, with a sample size of ~220 and an alpha of 0.05, our study would have a power of at least 80% to detect our pre-specified endpoints at 1 and 2 years.

For *Studies 1 and 2*, choroidal, retinal and RNFL thicknesses at all locations were compared between groups by two-way ANOVA with Tukey correction for multiple comparisons. Macular volume was compared using one-way ANOVA. Stepwise linear regression was used for multivariable analysis of variables showing linear relationship with choroidal thickness. For *Study 3*, within group thicknesses at all locations and macular volumes were compared at each timepoint by two-way ANOVA with Tukey correction for multiple comparisons or a mixed effects analysis where appropriate. Statistical analysis was performed using Prism version 8.2.1 (GraphPad Software Inc) and R version 3.6.1 (R Foundation). Data are presented as mean ± standard deviation. Parametric and non-parametric tests were used as appropriate to compare the differences between groups. For *Study 4*, unadjusted, univariable logistic regression models were used to examine the simple relationships between OCT metrics (macular volume (as a composite of retinal thickness) and choroidal thickness) and the pre-specified study outcomes. Then, multivariable logistic regression models were constructed to further investigate the relationship between each OCT metric and both study outcomes. These models were adjusted for important confounders which were identified a priori, and included: patient age, sex, baseline eGFR, systolic BP and urine protein:creatinine ratio. Confounding variables were added sequentially; the relative performance of each model was assessed by examining the likelihood ratio test statistic, whilst the goodness-of-fit of each model was evaluated by reviewing its associated Akaike Information Criterion (AIC) value. The variance inflation factor (VIF) was used to evaluate the degree of multicollinearity present in each model. A VIF of ≥2 was considered significant. The results of both univariable and multivariable logistic regression models were presented as odds ratios (ORs) and 95% confidence intervals (CIs).

### Reporting summary

Further information on research design is available in the Nature Portfolio Reporting Summary linked to this article.

## Data availability

Source data are provided with this paper. De-identified individual participant data are available from corresponding author (bean.dhaun@ed.ac.uk) for up to 6 months from online publication date. Source data are provided with this paper.

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

## Acknowledgements

We are very grateful to Nikolaos Tzoumas, Armin Saberi-Oskooi, Mike Aluwini, Inge Wolterink, Carla Gardiner, Richard Sandifort, Iris Nap, Judith Brouwer, Jamie Donachie, Loic Hayois, Ashley Meikle, Sophie Vennard, Jack McLachlan, Catherine Scriven, Zagros Kluth, and Caspian Kluth for their assistance and hard work during these studies. This research was funded by a project grant from Kidney Research UK (KRUK, KS_RP_003_20210111), a Research Excellence Award (RE/18/5/34216) from the British Heart Foundation (BHF) and support from the Edinburgh High Blood Pressure Foundation. T.F. was supported by a Clinical Research Training Fellowship from the Medical Research Council (MR/R017840/1). F.A.C. was supported by a Clinical Research Training Fellowship from KRUK (TF_006_20171124). D.P. was funded by a Chief Scientist Office PhD Research Fellowship (CAF/19/01). P.J.G. was supported by a BHF Clinical Research Training Fellowship (FS/CRTF/20/24079). N.D. was supported by a Senior Clinical Research Fellowship from the Chief Scientist Office (SCAF/19/02).

## Author contributions

N.D., D.J.W., B.D. and T.E.F. designed the study; T.E.F., D.P., F.C., E.G., C.B. and P.J.G. carried out the study and analysed data. All authors including G.C.O., J.W.D. and M.A.B. contributed to interpretation of data, drafting and revision of the final manuscript.

## Competing interests

The authors declare no competing interests.
