## [Peer Review File · Nature Communications]

Choroidal and retinal thinning in chronic kidney disease independently associate with eGFR decline and are modifiable with treatmentREVIEWER COMMENTS

Reviewer #1 (Remarks to the Author):

CKD is strongly associated with incident cardiovascular disease and CDK patients are more likely to die from cardiovascular disease than develop kidney failure, likely because of systemic vascular atrophy. eGFR is a key marker of CKD, but it is insensitive as it only becomes visibly abnormal once 50% of kidney is lost to fibrosis and atrophy. Kidney biopsies can be more reliable but are unpracticable in clinical practice. The authors of this paper had previously shown that CDK patients are characterized by choroidal thinning and macular volume loss in OCT, but the clinical significance and usefulness was unclear. Here they conducted several prospective studies using CDK patient cohorts that demonstrate a surprisingly tight relationship of OCT measures with CDK not only during progression but also in response to treatment. They demonstrate that choroidal macular volume and choroidal thickness (but not the NFL thickness) are decreased in CDK patients. In CDK patients with kidney transplants the choroidal thinning, and to a lesser degree the macular volume loss, is reversed. Remarkably, in CDK patients followed pre- and post- kidney transplantation the choroidal and retinal thickness very significantly re-augmented. Reversely the kidney donors (characterized by an increase of the eGFR) a gradual long-term loss of thickness and volume was observed. Last but not least in CDK patients thinned retina and choroid predicted the future eGFR development. There was a direct relationship of kidney fibrosis/CDK severity/lower eGFR and the OCT measures independently of age. This is a fascinating series of several clinical studies making use of CDK patients, kidney transplants, but also kidney donors to unravel the usefulness of macular volume and choroidal thickness measurements as proxies for the state of vascular fibrosis/constrictions elsewhere in the body. The study design is very elegant, in particular the longitudinal studies before/after kidney transplant recipients and donors. In the future, OCT measurements might be a useful, cheap, non-invasive clinical marker for CDK progression, and risk of cardiovascular events in CDK patients. Beyond CDK one wonders whether OCT measurements could have predictive use in kidney-independent cardiovascular diseases. Things that could be improved:

Choroidal volume: As pointed out by the authors, sub-macular choroidal volume might be a more accurate measure than choroidal thickness. As the authors have scanned the whole

macular area for macular volume calculations, wouldn't it be possible to calculate the choroidal volume and be even more accurate?

Macular volume: In most ocular diseases, an increase of macular volume is not usually a good sign as it is most often due to edema (wet AMD, DME). I guess we are talking here about a more discrete increases in volume? this should be better explained. Is there any indication whether this increase in volume is to intra- or inter-cellular volume differences? There is a puzzling difference in the OR (2,4) of macular volume for eGFR prediction compared to choroidal thickness (OR 1). The authors should address possible reasons for this.

Reviewer #2 (Remarks to the Author):

The authors show novel data linking non-invasive examination of the retina/choroid with OCT and kidney disease with 4 different clinical studies including healthy volunteers, CKD patients, kidney transplant recipients and kidney donors.

In these different clinical studies they show an association between choroidal/retinal thickness and kidney function. They suggest that OCT and retinal/choroid examination could represent a novel marker for kidney function and for assessing kidney prognosis.

In a more and more specialized medicine it is uncommon to see research works connecting two different medical fields. Here the authors provide novel and worthy data for chronic kidney disease patients. The group already published several good quality previous manuscripts on that topic.

However, I think that some points should be clarify in order to ameliorate the manuscript.

1/Introduction: the authors should explain briefly for a general reader which key aspects make the eye and the kidney "similar structures" as stated in the introduction.

2/ Population:

Overall, the reason for using these 4 populations should be better explained. The manuscript is composed by 4 studies and it could be confusing for the reader if the reasons for these 4 studies are not clearly stated.

The exclusion/inclusion criteria should be specified.

a. Why dialysis patients have been excluded from the studies? Is the retinal/choroidal

thinning in stage 5 CKD patients maximal?

b. Diabetes mellitus is one main cause of CKD and has an impact on retina/choroid. The authors should explain in the limitations how that exclusion may decrease the interpretability of the results in a general CKD population in a Western country.

3/ Who is performing OCT examination? Could it be done by a nephrologist or a technician in order to generalize the technique among CKD patients? What is the cost of OCT compared to currently used blood kidney function biomarkers?

4/ Steroids: a choroid thickening is observed after kidney transplantation. Many drugs as steroids are used during that period. The authors discuss tacrolimus effects measured with OCT. However, it would also be interesting to know how the steroids affect choroidal thickness?

5/OCT and living donors: I think that section should be toned down as the results show only trends (trend to a choroidal thinning after donation) and so the conclusions related to kidney donation side effects seem to be exaggerated and too alarming for the global nephrology community.

6/ Kidney transplant recipients follow-up: that is a very interesting study. Is the center performing graft protocol biopsies (month 3 or year 1 post-transplant) in order to correlate histological findings to OCT measurements? Is there any effect of acute graft rejection episodes on OCT findings?

7/ Only some choroidal locations thickness measurements are associated with kidney function in multivariate analysis. Could the authors explain the differences between choroidal locations 3 vs 2? Are these measurements standardized? What could be the mechanistic explanation for these findings?

8/ How kidney fibrosis is assessed in kidney biopsies?

Minor:

1/Page 14: I think the use of "trends" can be misleading for the reader in the Results section and only significant results should be emphasized (see section on kidney living donors for example). In my opinion trends can be discussed in "Discussion" section.

2/ Suppl Table 1 : Surprisingly creatinine levels are lacking in 9 healthy volunteers. Is it a mistake? Also, 1 healthy volunteer appears to have a GFR lower than 60 ml/min?

Reviewer #3 (Remarks to the Author):

Overall comments

The paper presents results from several clinical studies in the field of kidney disease. The background & rationale of the studies, as well as the clinical need addressed, are well explained and of high relevance. The study team should be praised for their research and efforts.

The authors chose to report several studies at once. Although I appreciate the study's common investigation team, background, recruitment, & clinical relevance of conclusions, I found this choice off-putting. It makes making sense of individual study's populations, methods, and statistical analysis harder, and at times confusing.

I strongly suggest restructuring each of the section mentioned above into clearer subsections, with sub-headings identifying each study. Enumerating them a), b), c)... as was done in figure 1 will make reading the paper substantially easier.

Trial conduct is good. Data collection is appropriate & well reported. I hope the anonymous data is made publically available for reproducibility. Tables & Figures are very clear and attractive, add interest and clarity to the findings. Well done.

I cannot comment on the clinical aspects of the paper, being outside my field of expertise.

I recommend publication, but with substantial revision of the Methods & Results sections, and the corrections I propose.

I hope my suggestions are helpful for the authors.

Title

The title brings attention to the main conclusion of the trials. My preference is to state type of trial & disease context, but this is down to editorial guidelines or personal choice.

Abstract

The abstract is clear, properly structured & exhaustive.

Methods

I recommend that paragraphs "Study design" and the first part of "Study details" be merged as concisely as possible into one section & placed this at the beginning of the "Methods" section, titled "Study design". This will increase clarity. Description of trial design (cross section, longitudinal) including allocation ratio belong here. As well as recruitment figures per study. Please state clearly how many longitudinal studies were conducted in main text (I struggled to figure out).

Primary & secondary endpoints should be stated more clearly, and explicitly labelled so. Each study should have one primary endpoint, and this is the same endpoint for which the study was powered. For instance, replace sentence "Our cross-sectional study assessed differences in OCT metrics between subject groups" with "The primary endpoint of the cross-sectional study was to assess differences in OCT metrics between subject groups", add definition of groups compared. Likewise "The primary endpoint of the longitudinal studies was to assess the differences in OCT metrics" (with description of groups compared).

Add a list of secondary endpoints per study. *IF* any secondary endpoints were not pre-planned (details not given, and should be) these should be labelled "exploratory" endpoints & a strong rationale be given for why these further endpoints were added. Any statistical inference performed on such endpoints is hypothesis generating & should not be reported as conclusive ("we have shown that" etc), you merely observed a trend worthy of further investigation.

Rename section "Subjects" to "Study populations". Eligibility/Ineligibility criteria are stated clearly.

Rename "Study power" to "Sample size determination" & place after "Study populations".

Section "OCT imaging" and the second part of "Study details" belong together, these describe study outcomes & study assessments (how the outcome were measured). Consider merging under one section named "Assessments". The extensive detail given is clearly clinically relevant, however some of the detail of how routine measurements were obtained belong to a protocol, and maybe could have been omitted unless unusual. They do no harm

however.

Statistical Analysis

Methods for statistical analysis are clearly stated. Consider stating which analyses are for primary endpoint vs secondary endpoint, at present it is inferred but always best to be clear.

Details on software used helps with reproducibility, but are the authors making a protocol, statistical analysis plan and the data set publicly available? If so, state it, with details.

"Available on request" is acceptable if a public repository (or journal website) are not used.

"Significance was set at $p < 0.05$ with Pearson's or Spearman's correlation coefficients used as appropriate". To me this is an odd statement. For the primary endpoint a threshold for statistical significance of $p < 0.05$ is clearly implied from how you powered the studies. For secondary endpoints, yes, you can state that you consider a $p < 0.05$ to detect a statistical significance between groups, however this inference is less robust & hypothesis generating rather than conclusive. I would remove the sentence, and in the results section be more careful with interpretation of $p < 0.05$. Do it sparingly.

$p < 0.05$ with power of 80% implies you reject the null hypothesis, i.e. there being no difference between the populations from which your samples are drawn at random. You have 80% confidence that this is correct when there is actually a difference, so 20% chance of Type 2 error (false negative). The p-value quantifies your Type 1 error (false positive) i.e. the chance you detected a difference when there is none. Any statistical significance should be interpreted this way. Thus talk of $p\text{-value} < 0.05$ showing "a significance differences between groups" is technically incorrect. You are inferring there's an actual difference between populations, based on the observed difference between study groups (samples). Difference between groups is given by the descriptive measures you observe (mean, se, confidence interval etc) and these should be reported under "Results" for each metric described therein.

Section "Association of OCT metrics with eGFR decline in patients with pre-dialysis CKD", first part, states the primary endpoint for which study? I concluded "a decline in eGFR" is

one of the OCT metrics, I am no expert in the field and hope this is clear to the readership. Otherwise consider adding this detail. This section seems to have a description of its own study assessments, justification of choice of endpoints, power calculation and statistical analysis. I find this very unhelpful and confusion. Either it presents a different study altogether and all this information is repetition and should be moved to its appropriate section, or it refers to one of the studies presented above. Is this an additional study? conducted on the same population, just considering a different endpoint? Make it clear.

Results

Why state recruitment here? This belongs to study design or study populations. Under results you should report the numbers actually analysed, might be the same as recruited, in which case say so and change the sentence from "25 patients ... were recruited" to "x patients ... were analysed" with numbers per group. Any big discrepancy between total recruited & total analysed should be made clear & justified.

Make clear which descriptive statistics describe the baseline characteristics of the populations (include in a table) vs the study endpoints. Ensure that all baseline variables used in analyses are included in tables. Add ranges (min, max) & present median (IQR) for skewed data. Comment if there's a notable difference between groups, this should go in Limitations if not expected.

Ensure all descriptives, not just some, are presented per group with absolute values.

Sentence "The retina was thinner in patients with CKD compared to healthy volunteers and this thinning was particularly apparent in the central retina (~5% thinner compared to health)" should give absolute values.

Present results explicitly identifying the primary endpoints for each study. For these you can make inference, i.e. say the study indicates a difference between populations, but refrain from interpretation (leave this for discussion). Saying "interestingly" of a finding at this stage isn't appropriate. Results should be presented as objectively as possible, without narratives. Result is what you observed, not how you interpret it. Summarise concisely with no comments on expectation or impact.

I strongly recommend against using such phrases as "we showed a difference" or even worse "we show no difference". You can merely say you weren't able to show any difference, not that there isn't one between groups based solely on $p < 0.05$. Too many p-values are meaningless anyway, suffering at the very least with problems of multiple comparison, and should only be reported for primary endpoint with any confidence. Anything else might be of interest (and hypothesis generating) but not conclusive. Especially if further, exploratory analyses, were conducted on the strength of previous results. You did not employ a hierarchical approach for such analyses, thus you are using the same sample more than once. This should be accounted for by reducing the p value for each further comparison, and you didn't. This is a limitation of some of your results and you should state so in "Limitations".

Discussion

This section is too long. Occasionally repeats information already supplied in methods. Suffers with too much unjustified inference. Consider revision & shortening. Clinical discussion on impact and context is relevant, but not my field of expertise thus I won't comment on content.

Limitations

Remove sentence "Thus, our methodology is robust, and our data are reproducible" both are matter of opinion which should be left to the reader having assess the limitations you disclose. You mean "results" rather than "data" anyway.

Also remove from this section any presentation of results/conclusions (e.g. "We found no relationship between the timing of OCT and any OCT metric. We have also demonstrated that there is no significant variability in metrics obtained when OCT scans are performed by different trained operators"). These are not limitations. Include only: limitation in trial design and conduct, sources of potential bias, imprecision, and the multiplicity of analyses. Comment on generalizability of results of bias in recruitment (for example, you recruited mostly white patients when the most likely affected people are from a different ethnicity).

Table 1

I don't see the point of this table in the main paper, move to supplementary. It shows

overall recruitment characteristics, but those are not as important as showing baseline characteristics split by group and study as in Tables 1 & 6.

Table 2

Report p-values to 3 df, for consistency.

Figure 1

Nice graphics. Use these same titles (including enumeration) in paper, for clarity.

Figure 2 & 3

My preference is not to include symbols denoting p-values in figures. Detracts from presentation of the data, which is the primary aim of the figure. p-values should go in appropriate tables instead. Follow journal guidelines, if any, on the matter. NEJM & JAMA don't like it either. Too many p-values overall detracts from credibility, suffering as they do from multiple comparisons & adequate cautiousness regarding their meaning.

If data doesn't show a particularly unusual distribution, why not present boxplots? Those are more informative. Panel A, C especially. This is relevant to all subsequent figures.

Supplementary Methods

Not sure why this is here? Sub-studies? Explain.

Supplementary Table 1, 6

Add totals.

Supplementary Table 2 & following tables

Be consistent with number of p-values df, use 3 across paper. Explicitly write trailing 0, i.e. 0.590 instead of 0.59.

Supplementary Table 8/9

Very busy tables. Here I would seriously consider dropping the p-values (and certainly don't highlight them in bold).

All supplementary figures

Definitely remove p-values. Again consider boxplots rather than scatter plots.

Supplementary Figure 9

Study flow diagram pretty unhelpful unless presents patient flow per study & per group. I appreciate you chose to report several studies in one paper, but usually this type of diagram is to support the comparisons you report in the paper, thus showing merely the totals is quite pointless. Yes, it does show when/where/why patients dropped out, and final analysis numbers & this adds information, but it again rises my reservation that several studies where presented all at once. If you can, split the diagram into separate panels, one per study reported.

Consider putting it in main paper, it goes well with Figure 1. Or at the very least the first of the supplementary figures. I think the main paper suffers for having one too many figures illustrating results. Just focus on only one per study. Thus you gain a space for the flow diagram.

REVIEWER 1

CKD is strongly associated with incident cardiovascular disease and CKD patients are more likely to die from cardiovascular disease than develop kidney failure, likely because of systemic vascular atrophy. eGFR is a key marker of CKD, but it is insensitive as it only becomes visibly abnormal once 50% of kidney is lost to fibrosis and atrophy. Kidney biopsies can be more reliable but are unpracticable in clinical practice. The authors of this paper had previously shown that CKD patients are characterized by choroidal thinning and macular volume loss in OCT, but the clinical significance and usefulness was unclear. Here, they conducted several prospective studies using CKD patient cohorts that demonstrate a surprisingly tight relationship of OCT measures with CKD not only during progression but also in response to treatment. They demonstrate that choroidal macular volume and choroidal thickness (but not the NFL thickness) are decreased in CKD patients. In CKD patients with kidney transplants the choroidal thinning, and to a lesser degree the macular volume loss, is reversed. Remarkably, in CKD patients followed pre- and post- kidney transplantation the choroidal and retinal thickness very significantly re-augmented. Conversely the kidney donors (characterized by an increase of the eGFR) a gradual long-term loss of thickness and volume was observed. Finally, in CKD patients thinned retina and choroid predicted the future eGFR development. There was a direct relationship of kidney fibrosis/CKD severity/lower eGFR and the OCT measures independently of age.

This is a fascinating series of several clinical studies making use of CKD patients, kidney transplants, but also kidney donors to unravel the usefulness of macular volume and choroidal thickness measurements as proxies for the state of vascular fibrosis/constrictions elsewhere in the body. The study design is very elegant, in particular the longitudinal studies before/after kidney transplant recipients and donors. In the future, OCT measurements might be a useful, cheap, non-invasive clinical marker for CKD progression, and risk of cardiovascular events in CKD patients. Beyond CKD one wonders whether OCT measurements could have predictive use in kidney-independent cardiovascular diseases.

We thank the Reviewer for recognising our work as a *'fascinating series of several clinical studies'* with a *'very elegant'* design. We hope our findings demonstrate the potential utility of OCT to act as a quick, relatively inexpensive, and non-invasive biomarker of kidney and cardiovascular disease. We also agree with the Reviewer that there is potential for OCT to provide insights in diseases other than those involving the kidney. Indeed, we have shown previously that there may be a role for OCT in the context of liver transplantation.¹

Specific comments

1. Choroidal volume: As pointed out by the authors, sub-macular choroidal volume might be a more accurate measure than choroidal thickness. As the authors have scanned the whole macular area for macular volume calculations, wouldn't it be possible to calculate the choroidal volume and be even more accurate?

Thank you for this very interesting suggestion. To our knowledge, there are no OCT devices that currently automatically segment and calculate choroidal thickness, area, or volume, and thus some degree of manual measurement is required. Measuring choroidal thickness on a single horizontal line scan image is quick, uses integrated calibrated callipers and, as we show here, is highly reproducible. Manually estimating choroidal area is significantly more labour intensive and error prone as it requires use of a freehand lasso tool to delineate the region of interest. Repeating this measurement for each of the 64 horizontal images that cover the macula region would allow estimation of choroidal volume but would vastly increase the data extraction time particularly in a study of several hundred patients.

Automated choroidal area and volume measurement tools have been developed by other groups² but these still require significant manual correction and are not universally transferrable across OCT platforms which often have different proprietary imaging data storage protocols. It is also worth bearing in mind that estimated volumes are derived from a complete 'set' of images that span the macular area thus if one or two images are not assessable (due to an inability in identifying layer

boundaries as result of poor image quality or alignment for example), the accuracy of volume calculations is significantly affected. This is particularly true of the deeply situated choroid more so than the superficial retina. A single high-quality cross-sectional image of a particular region of the choroid or retina is much easier to consistently acquire, reduces the risk of data loss and increases fidelity. Nevertheless, we are working with experts in the field of artificial intelligence to automate our choroidal thickness measures which in time may allow us to automate choroidal area and volume measures.

2. Macular volume: In most ocular diseases, an increase of macular volume is not usually a good sign as it is most often due to edema (wet AMD, DME). I guess we are talking here about a more discrete increase in volume? This should be better explained.

We thank the Reviewer for raising this important point. Macular volume may be increased by either focal accumulation of intra-retinal fluid, typically from local vascular leakage as exemplified in 'wet' age-related macular degeneration and diabetic macular oedema, or due to a generalised increase in retinal thickness across the whole of the macula. In the former scenario which is more common, scrutiny of the OCT image and thickness map would reveal such fluid collections; however, we did not identify any cases of this during our protocolised image quality assessment and analysis.

Our data suggest a generalised increase in retinal thickness (**Supplementary figures 7A & 7B**) as an explanation for the increase in macular volume which appears more evident in the inner (or central) macular zones. In addition, the association between an increase in macular volume and adverse ocular outcomes is probably because it indicates an underlying vasculopathy and the increase represents a deviation *away* from a 'healthy' or 'normal' macular volume derived from a control population. The ~2% increase in macular volume we have observed following kidney transplantation represents a deviation *towards* that of healthy controls. The implications of such increases for the ocular health of kidney transplant recipients are not clear and beyond the scope of our study but will be an important part of future work as ocular disease is common in these patients.³ Our revised manuscript includes a brief discussion of these points.

3. Is there any indication whether this increase in volume is due to intra- or inter-cellular volume differences?

The resolution of current OCT devices is insufficient to distinguish between changes in extra- and intracellular, or indeed extra- and intravascular compartments. Given the retina is comprised mainly of axons and neuronal cell bodies, thinning here is generally considered to represent neuronal loss. This is believed to be irreversible and certainly very unlikely to be reversed within the time course of our studies. The choroid is an almost entirely a vascular layer comprising small arteries, arterioles, and a dense fenestrated capillary network.⁴ Thus, the temporal changes we have observed here following kidney transplantation and kidney donation may represent changes in vascular tone. It is also possible that increases in choroidal thickness represent accumulation of extravascular fluid from fenestrated capillaries following fluid supplementation during the peri- and postoperative periods. However, the observation that choroidal thickness continues to increase up to a month after transplantation, well after fluid shifts have resolved, argues against this as significant contributor to our results.

The retina also contains two discrete capillary plexi which are particularly dense around the inner parafoveal macula. Given the changes in the choroid, it is plausible that the small change in global macular volume represents vasodilatation and increased perfusion through these vascular plexi. In support of this, we did note that the parafoveal regions had the largest increases in macular volume (**Supplementary figure 7B**). Improved perfusion of the retina may be beneficial in preventing further neuronal ischaemia and neovascularisation but is unlikely to promote regeneration. Our cross-sectional data support this concept by showing that compared to matched controls, kidney transplant recipients had persistent macular thinning despite similar choroidal thickness.

4. There is a puzzling difference in the OR (2,4) of macular volume for eGFR prediction compared to choroidal thickness (OR 1). The authors should address possible reasons for this.

The stronger independent association observed between reduction in macular volume and estimated glomerular filtration rate (eGFR) decline than that seen with choroidal thinning could be due to several reasons. The vascular nature of the choroid means that choroidal thickness is potentially subject to influence by factors such as blood pressure, sympathetic nervous system activation, volume status and vasoactive medications. Ocular factors such as axial length, which we did not measure, can also influence choroidal thickness but not macular volume or retinal thickness. These factors serve to increase the heterogeneity of choroidal thickness in any given population, and it would be impractical to consider and adjust for of all these in our analyses given our study size. Despite this, we still found an important association between choroidal thickness and eGFR decline which strengthens the utility of choroidal thickness as a biomarker for adverse kidney disease outcomes. An interesting extension of choroidal thickness could be examining the rate of choroidal thinning over time as a marker of increased risk of future kidney function decline. We have added a discussion of these points to our revised manuscript.

REVIEWER 2

The authors show novel data linking non-invasive examination of the retina/choroid with OCT and kidney disease with 4 different clinical studies including healthy volunteers, CKD patients, kidney transplant recipients and kidney donors. In these different clinical studies, they show an association between choroidal/retinal thickness and kidney function. They suggest that OCT and retinal/choroid examination could represent a novel marker for kidney function and for assessing kidney prognosis. In a more and more specialized medicine, it is uncommon to see research works connecting two different medical fields. Here the authors provide novel and worthy data for chronic kidney disease patients. The group already published several good quality previous manuscripts on that topic. However, I think that some points should be clarified to ameliorate the manuscript.

We thank the Reviewer for their positive comments and the recognition that our work spans more than one medical field. We provide responses to their queries below.

Major comments

1. Introduction: the authors should explain briefly for a general reader which key aspects make the eye and the kidney 'similar structures' as stated in the introduction.

We thank the Reviewer for this suggestion. We have revised our manuscript accordingly.

'Bruch's membrane in the eye and the glomerular basement membrane (GBM) both contain a network of $\alpha 3$, $\alpha 4$ and $\alpha 5$ type IV collagen chains.^{5,6} Thus, inherited diseases of type IV collagen manifest with co-existent nephropathy and retinopathy as seen in Alport syndrome.⁷ The microcirculation of the retina can be subdivided into retinal and choroidal circulations. The choroidal capillary (choriocapillaris) endothelium has ~80 nm fenestrations allowing fluid exchange within the subretinal space⁸ similar to the glomerular endothelium for ultrafiltration into the urinary space.⁹ In addition, the eye and kidney have matched chorioretinal and corticomedullary oxygen gradients, respectively, and excessive activation of the renin-angiotensin-aldosterone and endothelin systems are implicated in the development and progression of both retinopathy and CKD.¹⁰⁻¹³ The choroidal circulation receives ~80% of ocular blood flow and passively oxygenates key visual apparatus including the pigment epithelium and photoreceptors particularly within the avascular fovea, suggesting a critical role in maintaining global retinal health.⁴ Choroidal vascular change may therefore predate the onset of overt retinopathy and, if detectable, might allow earlier identification of incipient disease..

2. Population. Overall, the reason for using these 4 populations should be better explained. The manuscript is composed by 4 studies, and it could be confusing for the reader if the reasons for these 4 studies are not clearly stated.

We thank the Reviewer for this suggestion to improve the clarity of the manuscript. We chose to examine patients with chronic kidney disease (CKD) because of the similarities between the eye and the kidney as outlined above, and because there is a clear unmet need for reliable biomarkers that identify kidney injury, track treatment response, and predict longer-term outcomes. We first studied patients with CKD to assess, in a larger cohort than in our previous work,¹⁴ if OCT metrics could reflect kidney damage compared to healthy controls. We also studied patients with a functional kidney transplant to see if OCT metrics differed in this subgroup of CKD who had received what is considered the optimal treatment for kidney failure, namely a transplant. We then took advantage of two commonly occurring clinical scenarios, specifically, living donor kidney transplantation and donor nephrectomy, to study the effects of an acute gain or loss of kidney function on OCT metrics to assess if these could serve as a dynamic biomarker to track damage and response to treatment. Our final study enrolled patients with CKD who were at risk of kidney disease progression to examine whether OCT metrics could be used to identify those at greatest risk and, if so, allow this group to be targeted with additional renoprotective (and cardioprotective) treatments. We hope this clarifies the aims of our studies for the Reviewer.

3. The exclusion/inclusion criteria should be specified.

a) Why dialysis patients have been excluded from the studies? Is the retinal/choroidal thinning in stage 5 CKD patients maximal?

Thank you for this comment. We excluded dialysis patients from our cross-sectional study as they comprise a relatively small proportion of patients with CKD and represent the endpoint of kidney damage for which the optimal treatment for improving outcomes is kidney transplantation. These patients also exhibit a unique phenotype of systemic vasculopathy and end organ damage, derived in part from the nature of dialysis itself which may confound changes in choroid and retina. The timing (both pre/post or after the two-day dialysis break, with respect to those receiving maintenance haemodialysis) and intensity of dialysis could also introduce confounding in cross-sectional studies. Our concerns are supported by existing studies of OCT metrics in haemodialysis patients (summarised in 2020 *Kidney International* publication¹⁵) which generally, but not universally, report acute reductions of retinal and choroidal thickness that associate with degree of ultrafiltration and solute removal. Studies of dialysis patients using the same OCT device as our study,^{16,17} do show a markedly reduced macular volume and choroidal thickness compared to the values we observed in pre-dialysis CKD patients (macular volume: 7.9 mm³ versus 8.4 mm³; subfoveal choroidal thickness: 233 µm versus 294µm), in keeping with advanced systemic end organ damage typical of dialysis patients.

b) Diabetes mellitus is one main causes of CKD and has an impact on retina/choroid. The authors should explain in the limitations how that exclusion may decrease the interpretability of the results in a general CKD population in a Western country.

We thank the Reviewer for raising this excellent point. We excluded patients with diabetes mellitus as the presence of concomitant diabetic eye disease (retinopathy, maculopathy and choroidopathy) would confound interpretation of changes related to kidney disease *per se*. Indeed, studies have found choroidal thickening in diabetics even in the absence of overt retinopathy.¹⁸ Additionally, treatments for diabetic eye disease such as pan-retinal laser photocoagulation¹⁹ and anti-vascular endothelial growth factor therapy²⁰ directly affect retinal and choroidal thickness and so are an additional source of confounding. We have revised the limitations sections of the manuscript to include these points.

4. Who is performing OCT examination? Could it be done by a nephrologist or a technician in order to generalize the technique among CKD patients? What is the cost of OCT compared to currently used blood kidney function biomarkers?

In our studies, OCT imaging was performed by nephrologists who underwent training in OCT image acquisition and analysis from specialist Heidelberg Engineering technicians. The technique is simple to learn and easily transferrable to clinical practice and has become standard care in the ophthalmology clinics and in many commercial opticians. A precise cost for each OCT scan is challenging; however, the device used in our study has a list price of £42,000 GBP (~\$55,000 USD) and commercial opticians offer an OCT scan for £10 GBP (\$13 USD). Thus, the cost per scan in a publicly funded healthcare system is likely to be less. This compares favourably with a £4 GBP (\$5 USD) cost associated with a blood test for creatinine.

5. Steroids: a choroid thickening is observed after kidney transplantation. Many drugs as steroids are used during that period. The authors discuss tacrolimus effects measured with OCT. However, it would also be interesting to know how the steroids affect choroidal thickness?

We agree with Reviewer that medication use, including systemic glucocorticoids, could contribute to our observations. We choose to explore the potential influence of tacrolimus in detail because of its known vasculo- and neurotoxic effects²¹ as well as the potential to quantitatively estimate exposure *via* therapeutic monitoring of plasma drug levels. Our hospital does not routinely monitor mycophenolic acid levels and so assessing mycophenolate mofetil and glucocorticoid exposure is reliant on prescription recommendations which in an observational study does not allow reliable quantification of drug exposure.

It is possible that glucocorticoids may play a role in the acute increase in choroidal thickness in kidney transplant recipients, as these drugs have been shown to promote choroidal vasodilation in a rat model of central serous chorioretinopathy (CSCR).²² However, clinical data are inconclusive with observational studies reporting an association between glucocorticoids and the risk of CSCR²³ whereas longitudinal studies in non-transplant patients failed to demonstrate any change in choroidal thickness despite using similar or even greater doses of glucocorticoids than used in our standard kidney transplant immunosuppression protocol.^{24,25}

6. OCT and living donors: I think that section should be toned down as the results show only trends (trend to a choroidal thinning after donation) and so the conclusions related to kidney donation side effects seem to be exaggerated and too alarming for the global nephrology community.

Thank you for this comment. Study 3 investigated the impact of changing GFR on OCT metrics. To do we used two complementary paradigms. First, the receiving of a kidney transplant where the recipient experiences rapid improvements in GFR and the second, living kidney donation, where GFR is rapidly lost. Given the pre-defined study design and outcome we believe it is important to report the findings, even if non-significant. Additionally, it is now well-recognised that kidney donation is associated with important longer-term risks of developing both chronic kidney and cardiovascular disease.^{26,27} Currently, we are unable to identify kidney donors most at risk of these complications. Recently, we showed the potential utility of OCT-angiography (OCT-A) – which defines retinal vascular density and geometry – in discriminating kidney donors with normal kidney function from matched healthy volunteers.²⁸ The phenotype of the donors’ retinal vasculature evolved over time following donation to resemble that of patients with CKD. The current data suggest that OCT, which provides complementary data to OCT-A, *may* also have a role in long-term risk stratification of kidney donors, especially if future studies link OCT metrics to cardiovascular disease risk. We feel that our discussion gives a balanced view of our findings.

7. Kidney transplant recipients follow-up: that is a very interesting study. Is the center performing graft protocol biopsies (month 3 or year 1 post-transplant) in order to correlate histological findings to OCT measurements? Is there any effect of acute graft rejection episodes on OCT findings?

We thank the Reviewer for their positive comments. Our hospital does not routinely perform protocol graft biopsies so we are unable to systematically assess histological change in kidney transplants with contemporaneous OCT metrics. Two kidney transplant recipients enrolled in our longitudinal study did experience acute rejection in the early post-transplant period. Interestingly, we observed differences in the pattern of choroidal thickness change compared to each other and the overall group.

OCT & acute rejection in kidney transplant recipients.

Line graph showing plots of the median interval change in choroidal thickness from pre-transplant in all recipients (blue), subject 6 (red) and subject 14 (black). Compared to the median increase of the group, subject 6 had a comparatively blunted increase in choroidal thickness at one-week post-transplant in the context of slow graft function due acute rejection which improved markedly after pulsed glucocorticoids, mirroring improvement in graft function. Subject 14 developed new acute graft dysfunction approximately 3 months after transplantation. We observed a concomitant fall in choroidal thickness back to pre-transplant levels in the context of severe vascular rejection that did not improve despite pulsed glucocorticoids and anti-thymocyte globulin to stabilise graft function. TCMR: T-cell mediated rejection.

Whilst these are only two examples, they suggest the fascinating potential utility of OCT to non-invasively identify acute kidney damage and track treatment response. This will be the focus of future work in our OCT research programme.

8. Only some choroidal locations thickness measurements are associated with kidney function in multivariate analysis. Are these measurements standardized? Could the authors explain the differences between choroidal locations 3 versus 2? What could be the mechanistic explanation for these findings?

There are no standardised locations for measurement of choroidal thickness. Given the critical role of the choroid in passively oxygenating the avascular fovea, subfoveal choroidal thickness (which corresponds to our location II) has been the most widely assessed in examining links to eye disease. Anatomically the choroid is thickest in the subfoveal region to sustain the high metabolic requirement of the visual apparatus and photoreceptor population in this vital visual area.^{4,29} From this central location, the choroid thins as it extends superiorly/inferior and nasally/temporally to mirror the lower metabolic demands of the peripheral retina.^{4,29} Choosing locations a specific distance from the subfovea is arbitrary; however, we selected 2 mm nasal and temporal to match the regions included in the Early Treatment of Diabetic Retinopathy study map, a clinically relevant standard.³⁰

The difference in association between kidney function decline and choroidal thickness assessed at locations II and III may be due to several reasons. As outlined above, the choroid is thickest in the subfoveal (location II) region. Thus, identification of the posterior boundary of choroid, the choroidal-scleral junction, is most challenging at this location and thus measurements here may have greater variation which will weaken the strength of any potential associations. In addition, the same degree of thinning, whether functional or structural in aetiology, in response to a disease process may be more evident in peripheral choroidal locations which have lower baseline vascular density due to their inherent lower metabolic demands. Finally, the vascular trunks that supply the choriocapillaris are more numerous in the foveal region^{4,31} and therefore the vasculopathy associated with CKD may be evident earlier in the more peripheral choroidal regions and thus associate with risk of kidney function decline more readily than subfoveal regions.

9. How was kidney fibrosis assessed in kidney biopsies?

All kidney biopsies were reviewed by two experienced nephropathologists. Fibrosis was assessed on light microscopy on H&E-stained sections. The proportion of glomerulosclerosis and interstitial fibrosis and tubular atrophy (IFTA) were assessed across the whole biopsy specimen.

Minor comments

1. Page 14: I think the use of ‘trends’ can be misleading for the reader in the Results section and only significant results should be emphasized (see section on kidney living donors for example). In my opinion trends can be discussed in ‘Discussion’ section.

We appreciate the Reviewer’s comment. In general, we have only included positive and important negative results in our manuscript. However, we believe it is important to report one of our primary outcomes and highlight near statistical significance of these data. We have revised our manuscript to include trends in the discussion and qualify the use of ‘trends’ by including p values to illustrate this.

2. Supplementary table 1: Surprisingly creatinine levels are lacking in 9 healthy volunteers. Is it a mistake? Also, 1 healthy volunteer appears to have a GFR lower than 60 ml/min?

We thank the Reviewer for highlighting this transcription error and apologise for the mistake. We have corrected the table accordingly.

REVIEWER 3

The paper presents results from several clinical studies in the field of kidney disease. The background & rationale of the studies, as well as the clinical need addressed, are well explained and of high relevance. The study team should be praised for their research and efforts. The authors chose to report several studies at once. Although I appreciate the study's common investigation team, background, recruitment, & clinical relevance of conclusions, I found this choice off-putting. It makes making sense of individual study's populations, methods, and statistical analysis harder, and at times confusing. I strongly suggest restructuring each of the section mentioned above into clearer subsections, with sub-headings identifying each study. Enumerating them a), b), c)... as was done in figure 1 will make reading the paper substantially easier. Trial conduct is good. Data collection is appropriate & well reported. I hope the anonymous data is made publically available for reproducibility. Tables & Figures are very clear and attractive, add interest and clarity to the findings. Well done. I cannot comment on the clinical aspects of the paper, being outside my field of expertise.

I recommend publication, but with substantial revision of the Methods & Results sections, and the corrections I propose. I hope my suggestions are helpful for the authors.

We would like to thank the Reviewer for their very positive comments about our work and their recommendation that it should be published in *Nature Communications*. We have found the Reviewer's detailed comments very helpful in improving the clarity of the manuscript. We have incorporated many of their suggested changes. For any changes not incorporated, we have explained our rationale.

1. Title.

The title brings attention to the main conclusion of the trials. My preference is to state type of trial & disease context, but this is down to editorial guidelines or personal choice.

Thank you for the suggestion. We have deliberated over various potential titles, and we would be keen to keep the title as it is [*Choroidal & retinal thinning in chronic kidney disease are modifiable with treatment & independently associate with eGFR decline*] as it describes the main findings of the study.

2. Abstract.

The abstract is clear, properly structured & exhaustive.

Thank you.

3. Methods

I recommend that paragraphs "Study design" and the first part of "Study details" be merged as concisely as possible into one section & placed this at the beginning of the "Methods" section, titled "Study design". This will increase clarity. Description of trial design (cross section, longitudinal) including allocation ratio belong here. As well as recruitment figures per study. Please state clearly how many longitudinal studies were conducted in main text (I struggled to figure out).

Thank you for this suggestion. As suggested, we have merged the 'Study design' and 'Study details' sections in to one section under the heading 'Study design'.

Primary & secondary endpoints should be stated more clearly, and explicitly labelled so. Each study should have one primary endpoint, and this is the same endpoint for which the study was powered. For instance, replace sentence "Our cross-sectional study assessed differences in OCT metrics between subject groups" with "The primary endpoint of the cross-sectional study was to assess differences in OCT metrics between subject groups", add definition of groups compared. Likewise, "The primary endpoint of the longitudinal studies was to assess the differences in OCT metrics" (with description of groups compared).

Again, we appreciate this helpful suggestion from the Reviewer. We have modified the text to clearly state the primary and secondary endpoints for each study performed.

Add a list of secondary endpoints per study. *IF* any secondary endpoints were not pre-planned (details not given and should be) these should be labelled "exploratory" endpoints & a strong rationale be given for why these further endpoints were added. Any statistical inference performed on such endpoints is hypothesis generating & should not be reported as conclusive ("we have shown that" etc), you merely observed a trend worthy of further investigation.

A very helpful comment. The revised manuscript now describes the pre-planned secondary endpoints for each study as well as any exploratory endpoints that were not defined a priori.

Rename section "Subjects" to "Study populations". Eligibility/Ineligibility criteria are stated clearly. Rename "Study power" to "Sample size determination" & place after "Study populations".

Thank you. All of these suggestions have helped in revising the Methods section of the manuscript.

Section "OCT imaging" and the second part of "Study details" belong together, these describe study outcomes & study assessments (how the outcome were measured). Consider merging under one section named "Assessments". The extensive detail given is clearly clinically relevant, however some of the detail of how routine measurements were obtained belong to a protocol, and maybe could have been omitted unless unusual. They do no harm however.

As suggested, we have brought together the text previously placed under the 'OCT imaging' and (some) from the 'Study details' section under a new section 'Study assessments'. Given the measurements we describe are not usual, we have left them in the manuscript.

4. Statistical Analysis

Methods for statistical analysis are clearly stated. Consider stating which analyses are for primary endpoint vs secondary endpoint, at present it is inferred but always best to be clear. Details on software used helps with reproducibility, but are the authors making a protocol, statistical analysis plan and the data set publicly available? If so, state it, with details. "Available on request" is acceptable if a public repository (or journal website) are not used.

Thank you for these comments. We are not planning on making any protocol, statistical analysis plan or dataset publicly available separate to what is contained within the manuscript. As far as possible, we have revised the text to clearly state which analyses were used for which endpoints where not already clear.

"Significance was set at $p < 0.05$ with Pearson's or Spearman's correlation coefficients used as appropriate". To me this is an odd statement. For the primary endpoint a threshold for statistical significance of $p < 0.05$ is clearly implied from how you powered the studies. For secondary endpoints, yes, you can state that you consider a $p < 0.05$ to detect a statistical significance between groups, however this inference is less robust & hypothesis generating rather than conclusive. I would remove the sentence, and in the results section be more careful with interpretation of $p < 0.05$. Do it sparingly.

Thank you for this comment. We have removed this sentence from the revised 'Statistical analysis section'.

$p < 0.05$ with power of 80% implies you reject the null hypothesis, i.e. there being no difference between the populations from which your samples are drawn at random. You have 80% confidence that this is correct when there is actually a difference, so 20% chance of Type 2

error (false negative). The p-value quantifies your Type 1 error (false positive) i.e. the chance you detected a difference when there is none. Any statistical significance should be interpreted this way. Thus, talk of $p\text{-value} < 0.05$ showing "a significance differences between groups" is technically incorrect. You are inferring there's an actual difference between populations, based on the observed difference between study groups (samples). Difference between groups is given by the descriptive measures you observe (mean, se, confidence interval etc) and these should be reported under "Results" for each metric described therein.

We appreciate the detailed comments by the Reviewer here. As far as possible we have tried to present the results with the relevant descriptive details whilst being conscious of the fact that we need to make this section accessible to the reader.

Section "Association of OCT metrics with eGFR decline in patients with pre-dialysis CKD", first part, states the primary endpoint for which study? I concluded "a decline in eGFR" is one of the OCT metrics, I am no expert in the field and hope this is clear to the readership. Otherwise consider adding this detail. This section seems to have a description of its own study assessments, justification of choice of endpoints, power calculation and statistical analysis. I find this very unhelpful and confusion. Either it presents a different study altogether and all this information is repetition and should be moved to its appropriate section, or it refers to one of the studies presented above. Is this an additional study? conducted on the same population, just considering a different endpoint? Make it clear.

Thank you for this suggestion and we apologise for the lack of clarity. The details of this study were intentionally kept separate to improve clarity for the readership; this is a data linkage study examining patient outcomes whereas the preceding studies are experimental studies assessing OCT metrics. We will defer to the Editor here and if they feel there is lack of clarity, we would be happy to revise this section.

5. Results

Why state recruitment here? This belongs to study design or study populations. Under results you should report the numbers actually analysed, might be the same as recruited, in which case say so and change the sentence from "25 patients ... were recruited" to "x patients ... were analysed" with numbers per group. Any big discrepancy between total recruited & total analysed should be made clear & justified.

It has always been our practice, and that of others, to report the number of subjects recruited to experimental medicine studies at the beginning of the Results section. We also report the number of subjects for whom data were available at the various timepoints at which OCT metrics were assessed (see Supplementary tables 6 and 8, and Supplementary figure 9).

Make clear which descriptive statistics describe the baseline characteristics of the populations (include in a table) vs the study endpoints. Ensure that all baseline variables used in analyses are included in tables. Add ranges (min, max) & present median (IQR) for skewed data. Comment if there's a notable difference between groups, this should go in Limitations if not expected.

Thank you for this suggestion. We have added these details to the relevant tables.

Ensure all descriptives, not just some, are presented per group with absolute values. Sentence "The retina was thinner in patients with CKD compared to healthy volunteers and this thinning was particularly apparent in the central retina (~5% thinner compared to health)" should give absolute values.

Thank you for this suggestion. There were several reasons why we did not include absolute values (\pm SD) for all of the descriptives; overall, this was to optimise the communication of our findings to as broad a scientific readership as possible:

- Given the large amount of data presented, we wanted to avoid making this section 'number-heavy' and inaccessible for the reader that it lost its scientific novelty;
- To avoid repetition as the tables and figures that we refer to throughout provide the actual absolute values (and where changes from baseline are given in the figures, we do provide baseline values);
- Given this is an entirely new area of research for many fields outside the eye, we wanted to provide the reader with context for the changes we saw (e.g., 'the choroid was thinned by ~10% compared to healthy subjects').

Present results explicitly identifying the primary endpoints for each study. For these you can make inference, i.e. say the study indicates a difference between populations, but refrain from interpretation (leave this for discussion). Saying "interestingly" of a finding at this stage isn't appropriate. Results should be presented as objectively as possible, without narratives. Result is what you observed, not how you interpret it. Surmise concisely with no comments on expectation or impact.

We have edited this section and removed and subjective comments on our findings.

I strongly recommend against using such phrases as "we showed a difference" or even worse "we show no difference". You can merely say you weren't able to show any difference, not that there isn't one between groups based solely on $p < 0.05$. Too many p-values are meaningless anyway, suffering at the very least with problems of multiple comparison, and should only be reported for primary endpoint with any confidence. Anything else might be of interest (and hypothesis generating) but not conclusive. Especially if further, exploratory analyses, were conducted on the strength of previous results. You did not employ a hierarchical approach for such analyses, thus you are using the same sample more than once. This should be accounted for by reducing the p value for each further comparison, and you didn't. This is a limitation of some of your results and you should state so in "Limitations".

Again, we appreciate the Reviewer's attention to detail, and we agree with their comments. We have tried to write this manuscript which presents very novel data that span several medical fields in a way that is both scientifically rigorous but also interesting and accessible to read. We apologise if, at times, our stylistic use of the English language has compromised the correct statistical communication of the data. We have tried to rectify this as much as possible.

6. Discussion

This section is too long. Occasionally repeats information already supplied in methods. Suffers with too much unjustified inference. Consider revision & shortening. Clinical discussion on impact and context is relevant, but not my field of expertise thus I won't comment on content.

Thank you for these comments. We have tried to make the Discussion relevant and concise. Indeed, it is similar in length or shorter than in other manuscripts published in the Journal. Given our manuscript covers several medical disciplines we feel this section needs to be broadly accessible.

7. Limitations

Remove sentence "Thus, our methodology is robust, and our data are reproducible" both are matter of opinion which should be left to the reader having assess the limitations you disclose. You mean "results" rather than "data" anyway.

As suggested, we have removed this sentence.

Also remove from this section any presentation of results/conclusions (e.g. "We found no relationship between the timing of OCT and any OCT metric. We have also demonstrated that there is no significant variability in metrics obtained when OCT scans are performed by different trained operators"). These are not limitations. Include only: limitation in trial design

and conduct, sources of potential bias, imprecision, and the multiplicity of analyses. Comment on generalizability of results of bias in recruitment (for example, you recruited mostly white patients when the most likely affected people are from a different ethnicity).

We have modified the Limitations section as suggested and removed any presentation of results and conclusions. This has also resulted in a shortening of the Discussion.

8. Table 1

I don't see the point of this table in the main paper, move to supplementary. It shows overall recruitment characteristics, but those are not as important as showing baseline characteristics split by group and study as in Tables 1 & 6.

As suggested, we have move Table 1 to the supplement (now Supplementary table xx).

9. Table 2

Report p-values to 3 df, for consistency.

Thank you for this suggestion. We have now reported all p values to 3 decimal places throughout the manuscript.

10. Figure 1

Nice graphics. Use these same titles (including enumeration) in paper, for clarity.

Thank you for the positive feedback! We have now included the Study numbers in the main Results section. We would prefer to keep the text in the subheadings here as it is given it better summarises the results for the respective section. The text in the Figure is abbreviated to avoid the Figure looking cluttered.

11. Figures 2 & 3

My preference is not to include symbols denoting p-values in figures. Detracts from presentation of the data, which is the primary aim of the figure. p-values should go in appropriate tables instead. Follow journal guidelines, if any, on the matter. NEJM & JAMA don't like it either. Too many p-values overall detracts from credibility, suffering as they do from multiple comparisons & adequate cautiousness regarding their meaning. If data doesn't show a particularly unusual distribution, why not present boxplots? Those are more informative. Panel A, C especially. This is relevant to all subsequent figures.

We appreciate the Reviewer's comments and suggestions. When originally writing this manuscript we did consider how best to present the data, and this included box plots. Given the nature of the data, we opted for the dot plots. Hopefully, as can be seen below, these provide a better visual presentation of the data. With regards to denoting p-values in these graphs, we would prefer to leave these in but will be guided by the Editors.

13. Supplementary Methods. Not sure why this is here? Sub-studies? Explain.

We have moved these from the supplement to the main Methods section.

14. Supplementary Table 1, 6. Add totals.

As suggested, we have added these totals in Supplementary tables 1 and 6.

15. Supplementary Table 2 & following tables

Be consistent with number of p-values df, use 3 across paper. Explicitly write trailing 0, i.e. 0.590 instead of 0.59.

Thank you. We have now consistently reported all p-values to 3 decimal places throughout the tables.

16. Supplementary Table 8/9

Very busy tables. Here I would seriously consider dropping the p-values (and certainly don't highlight them in bold).

We appreciate that these are busy tables, but we feel that including these data are important for the readers. Although we have kept the p-values, these are no longer in bold.

17. All supplementary figures

Again, consider boxplots rather than scatter plots. Definitely remove p-values.

As described in our response to the Reviewer's comment 11, we have left the data presented as dot plots rather than changing these to box plots. Also, given not all the data are described in the main text (e.g., the change in choroidal thickness from pre-transplant to each of the 5 timepoints in the post-transplant period, Supplementary figure 6), we feel it is important to show the p-values within the figures to allow the reader to appreciate the time course of the change.

18. Supplementary Figure 9

Study flow diagram pretty unhelpful unless presents patient flow per study & per group. I appreciate you chose to report several studies in one paper, but usually this type of diagram is to support the comparisons you report in the paper, thus showing merely the totals is quite pointless. Yes, it does show when/where/why patients dropped out, and final analysis numbers & this adds information, but it again rises my reservation that several studies were presented all at once. If you can, split the diagram into separate panels, one per study reported. Consider putting it in main paper, it goes well with Figure 1. Or at the very least the first of the supplementary figures. I think the main paper suffers for having one too many figures illustrating results. Just focus on only one per study. Thus, you gain a space for the flow diagram.

A CONSORT diagram, like that shown in Supplementary figure 9, is expected for data linkage studies of this type. We have extensive experience in these sorts of studies.³²⁻³⁵ We have also provided similar data, in terms of number of subjects and how these change over the follow-up period, for the other studies (e.g., see Supplementary figures 6 and 8). We would prefer to keep this figure in the manuscript and as Supplementary but again will be guided by the Editors. This is also in line with the Reviewer's suggestion for how we present our results (see point 5, comment 1).

References

1. Gifford FJ, Moroni F, Farrah TE, Hetherington K, MacGillivray TJ, Hayes PC, Dhaun N, Fallowfield JA. The Eye as a Non-Invasive Window to the Microcirculation in Liver Cirrhosis: A Prospective Pilot Study. *J Clin Med*. 2020;9. doi: 10.3390/jcm9103332
2. Chhablani J, Barteselli G, Wang H, El-Emam S, Kozak I, Doede AL, Bartsch DU, Cheng L, Freeman WR. Repeatability and reproducibility of manual choroidal volume measurements using enhanced depth imaging optical coherence tomography. *Investigative ophthalmology & visual science*. 2012;53:2274-2280. doi: 10.1167/iovs.12-9435
3. Berindán K, Nemes B, Szabó RP, Módis L, Jr. Ophthalmic Findings in Patients After Renal Transplantation. *Transplant Proc*. 2017;49:1526-1529. doi: 10.1016/j.transproceed.2017.06.016
4. Nickla DL, Wallman J. The multifunctional choroid. *Prog Retin Eye Res*. 2010;29:144-168. doi: 10.1016/j.preteyeres.2009.12.002
5. Booij JC, Baas DC, Beisekeeva J, Gorgels TGMF, Bergen AAB. The dynamic nature of Bruch's membrane. *Prog Retin Eye Res*. 2010;29:1-18. doi: <https://doi.org/10.1016/j.preteyeres.2009.08.003>
6. Boutaud A, Borza D-B, Bondar O, Gunwar S, Netzer K-O, Singh N, Ninomiya Y, Sado Y, Noelken ME, Hudson BG. Type IV collagen of the glomerular basement membrane. *J Biol Chem*. 2000;275:30716-30724. doi: 10.1074/jbc.M004569200
7. Savige J, Sheth S, Leys A, Nicholson A, Mack HG, Colville D. Ocular features in Alport's syndrome: pathogenesis and clinical significance. *Clinical journal of the American Society of Nephrology : CJASN*. 2015;10:703-709. doi: 10.2215/CJN.10581014
8. Anand-Apte B. HJG. Developmental anatomy of the retinal and choroidal vasculature. In: *Encyclopedia of the Eye*. Amsterdam: Elsevier; 2010.
9. Haraldsson B, Nystrom J, Deen WM. Properties of the glomerular barrier and mechanisms of proteinuria. *Physiological reviews*. 2008;88:451-487. doi: 10.1152/physrev.00055.2006
10. Mauer M, Zinman B, Gardiner R, Suissa S, Sinaiko A, Strand T, Drummond K, Donnelly S, Goodyer P, Gubler MC, et al. Renal and retinal effects of enalapril and losartan in type 1 diabetes. *N Engl J Med*. 2009;361:40-51. doi: 10.1056/NEJMoa0808400
11. Chou JC, Rollins SD, Ye M, Battle D, Fawzi AA. Endothelin receptor-A antagonist attenuates retinal vascular and neuroretinal pathology in diabetic mice. *Investigative ophthalmology & visual science*. 2014;55:2516-2525. doi: 10.1167/iovs.13-13676
12. Dhaun N, Goddard J, Webb DJ. The endothelin system and its antagonism in chronic kidney disease. *Journal of the American Society of Nephrology : JASN*. 2006;17:943-955.
13. Heerspink HJL, Parving H-H, Andress DL, Bakris G, Correa-Rotter R, Hou F-F, Kitzman DW, Kohan D, Makino H, McMurray JJV, et al. Atrasentan and renal events in patients with type 2 diabetes and chronic kidney disease (SONAR): a double-blind, randomised, placebo-controlled trial. *Lancet (London, England)*. 2019;10.1016/S0140-6736(1019)30772-X. doi: 10.1016/S0140-6736(19)30772-X
14. Balmforth C, van Bragt JJ, Ruijs T, Cameron JR, Kimmitt R, Moorhouse R, Czopek A, Hu MK, Gallacher PJ, Dear JW, et al. Chorioretinal thinning in chronic kidney disease links to inflammation and endothelial dysfunction. *JCI Insight*. 2016;1:e89173. doi: 10.1172/jci.insight.89173

15. Farrah TE, Dhillon B, Keane PA, Webb DJ, Dhaun N. The eye, the kidney, and cardiovascular disease: old concepts, better tools, and new horizons. *Kidney international*. 2020;98:323-342. doi: 10.1016/j.kint.2020.01.039
16. Ulas F, Dogan U, Keles A, Ertlav M, Tekce H, Celebi S. Evaluation of choroidal and retinal thickness measurements using optical coherence tomography in non-diabetic haemodialysis patients. *International ophthalmology*. 2013;33:533-539. doi: 10.1007/s10792-013-9740-8
17. Yang SJ, Han YH, Song GI, Lee CH, Sohn SW. Changes of choroidal thickness, intraocular pressure and other optical coherence tomographic parameters after haemodialysis. *Clinical & experimental optometry*. 2013;96:494-499. doi: 10.1111/cxo.12056
18. Tavares Ferreira J, Vicente A, Proenca R, Santos BO, Cunha JP, Alves M, Papoila AL, Abegao Pinto L. Choroidal thickness in diabetic patients without diabetic retinopathy. *Retina*. 2018;38:795-804. doi: 10.1097/iae.0000000000001582
19. Okamoto M, Matsuura T, Ogata N. Effects of panretinal photocoagulation on choroidal thickness and choroidal blood flow in patients with severe nonproliferative diabetic retinopathy. *Retina*. 2016;36:805-811. doi: 10.1097/iae.0000000000000800
20. Okamoto M, Yamashita M, Ogata N. Effects of intravitreal injection of ranibizumab on choroidal structure and blood flow in eyes with diabetic macular edema. *Graefes Arch Clin Exp Ophthalmol*. 2018;256:885-892. doi: 10.1007/s00417-018-3939-3
21. Azzi JR, Sayegh MH, Mallat SG. Calcineurin inhibitors: 40 years later, can't live without. *J Immunol*. 2013;191:5785-5791. doi: 10.4049/jimmunol.1390055
22. Zhao M, Célérier I, Bousquet E, Jeanny JC, Jonet L, Savoldelli M, Offret O, Curan A, Farman N, Jaisser F, et al. Mineralocorticoid receptor is involved in rat and human ocular chorioretinopathy. *J Clin Invest*. 2012;122:2672-2679. doi: 10.1172/jci61427
23. Carvalho-Recchia CA, Yannuzzi LA, Negrão S, Spaide RF, Freund KB, Rodriguez-Coleman H, Lenharo M, Iida T. Corticosteroids and central serous chorioretinopathy. *Ophthalmology*. 2002;109:1834-1837. doi: 10.1016/s0161-6420(02)01117-x
24. Han JM, Hwang JM, Kim JS, Park KH, Woo SJ. Changes in choroidal thickness after systemic administration of high-dose corticosteroids: a pilot study. *Invest Ophthalmol Vis Sci*. 2014;55:440-445. doi: 10.1167/iovs.13-12854
25. Lee JH, Lee JY, Ra H, Kang NY, Baek J. Choroidal changes in eyes treated with high-dose systemic corticosteroids for optic neuritis. *Int J Ophthalmol*. 2020;13:1430-1435. doi: 10.18240/ijo.2020.09.15
26. Muzaale AD, Massie AB, Wang MC, Montgomery RA, McBride MA, Wainright JL, Segev DL. Risk of end-stage renal disease following live kidney donation. *JAMA : the journal of the American Medical Association*. 2014;311:579-586. doi: 10.1001/jama.2013.285141
27. Mjoen G, Hallan S, Hartmann A, Foss A, Midtvedt K, Oyen O, Reisaeter A, Pfeffer P, Jenssen T, Leivestad T, et al. Long-term risks for kidney donors. *Kidney international*. 2014;86:162-167. doi: 10.1038/ki.2013.460
28. Giarratano Y, Pugh D, Farrah TE, Oniscu GC, MacGillivray TJ, Dhillon B, Dhaun N, Bernabeu MO. Novel retinal vascular phenotypes for the potential assessment of long-term risk in living kidney donors. *Kidney international*. 2022;102:661-665. doi: 10.1016/j.kint.2022.06.019

29. Margolis R, Spaide RF. A pilot study of enhanced depth imaging optical coherence tomography of the choroid in normal eyes. *American journal of ophthalmology*. 2009;147:811-815. doi: 10.1016/j.ajo.2008.12.008
30. Grading diabetic retinopathy from stereoscopic color fundus photographs--an extension of the modified Airlie House classification. ETDRS report number 10. Early Treatment Diabetic Retinopathy Study Research Group. *Ophthalmology*. 1991;98:786-806.
31. Hayreh SS. In vivo choroidal circulation and its watershed zones. *Eye*. 1990;4:273-289. doi: 10.1038/eye.1990.39
32. Miller-Hodges E, Anand A, Shah ASV, Chapman AR, Gallacher P, Lee KK, Farrah T, Halbesma N, Blackmur JP, Newby DE, et al. High-Sensitivity Cardiac Troponin and the Risk Stratification of Patients With Renal Impairment Presenting With Suspected Acute Coronary Syndrome. *Circulation*. 2018;137:425-435. doi: 10.1161/CIRCULATIONAHA.117.030320
33. Gallacher PJ, Miller-Hodges E, Shah ASV, Anand A, Dhaun N, Mills NL, High SI. Use of High-Sensitivity Cardiac Troponin in Patients With Kidney Impairment: A Randomized Clinical Trial. *JAMA Intern Med*. 2021. doi: 10.1001/jamainternmed.2021.1184
34. Gallacher PJ, Miller-Hodges E, Shah ASV, Farrah TE, Halbesma N, Blackmur JP, Chapman AR, Adamson PD, Anand A, Strachan FE, et al. High-sensitivity cardiac troponin and the diagnosis of myocardial infarction in patients with kidney impairment. *Kidney international*. 2022;102:149-159. doi: 10.1016/j.kint.2022.02.019
35. Lambourg EJ, Gallacher PJ, Hunter RW, Siddiqui M, Miller-Hodges E, Chalmers JD, Pugh D, Dhaun N, Bell S. Cardiovascular outcomes in patients with chronic kidney disease and COVID-19: a multi-regional data-linkage study. *Eur Respir J*. 2022;60. doi: 10.1183/13993003.03168-2021

REVIEWERS' COMMENTS

Reviewer #1 (Remarks to the Author):

The authors have taken into account all of my critiques and comments.

Reviewer #2 (Remarks to the Author):

The revised manuscript has been importantly improved. I am satisfied with the author's responses to my questions/issues raised in my initial review.

I do not have additional comments.

Reviewer #3 (Remarks to the Author):

I am happy that my comments and suggestions were useful to the authors. A commendable amount of effort and thoughtfulness went into editing the manuscript.

I've reviewed the changes. I am happy that most of my concerns have been well addressed. I note that both the methods and the tables/figures have been substantially improved. I think to great effect. The methods, in particular, are now much clearer and well-structured.

Regarding the use of dot pots vs boxplots, and the displaying of p-values on figures, I refer to the journal's editorial guidelines. I understand the preference of the authors even though I don't share it myself. In the case of the dot-plots, the plots do at least give a good visual clue of the sample size (and thus end up exposing where the smaller/larger sample could caution against over-interpretation of p-values).

Presentation of statistical inference has also improved. Adding greater clarity and objectiveness.

However, I would like the following line revisited (lines 227-231) it's actually gotten worse in the new version:

“We also observed trends towards increased odds of an eGFR decline of $\geq 20\%$ at two years for every 1 mm³ 228 decrease in macular volume (OR 2.42, 0.94 to 6.57; $p=0.07$) and every 10 μm decrease in choroidal thickness at locations I (OR 1.01, 0.96 to 1.05; $p=0.82$) and III (OR 1.04, 0.98 to 1.09; $p=0.19$) (Table 2) but these did not reach statistical significance.”

There’s no such thing as “trending towards” an estimand or “observing a trend” towards statistical significance. I am worried here you chose to comment because the p -value of the first OR ($p=0.07$) is quite close to “statistical significance” but not close enough to the “magical” 0.05 threshold. But rejecting a null-hypothesis with 7% type-1 error isn’t very different than rejecting it at 5%, if the evidence is strong and the effect very medically relevant. And a doubling of risk (OR 2.42) seems to me very medically relevant. The ‘evidence’ however isn’t all that strong given how wide the CI is. See below.

But you group this larger risk with two much lower ones (OR 10.1 and 1.04) which present for a statistician a very different strength of evidence in the inference (the confidence intervals and corresponding p -values) you observed three substantially different increases of risk: in the first case is more than doubling (OR 2.42 but with a CI going from 0.94, so in line with the other lower bounds, to a very high 6.57) and in the other two is very small (1 and 4% increases, CI quite tight around 1).

Without knowing much of the medical background, I would say a doubling of risk is likely to be worrying, while the other two effects are very much minor.

But looking at the confidence intervals there’s the higher OR having a wide confidence interval, so the evidence there for a definite effect is quite weak. Could be you need more data, or maybe something else. It’s hard to comment without seeing the source data. Maybe there’s similarity between the three effects and there’s an overall link. And maybe it’s just random noise in the data. At best you can generate an hypothesis of why this would warrant medical consideration, and what might cause the first variable to have quite a wider CI.

So, I would re-phrase simply saying:

“We additionally observed a large increase in the odds of an eGFR decline of $\geq 20\%$ at two

years for every 1 mm³ 228 decrease in macular volume (OR 2.42, 0.94 to 6.57; p =0.07) and more modest increases for every 10 μm decrease in choroidal thickness at locations I (OR 1.01, 0.96 to 1.05; p =0.82) and III (OR 1.04, 0.98 to 1.09; p =0.19) (Table 2).” Then formulate a medical hypothesis, what is of interest here? would you be very worried for a doubling of risk? so much so that you would monitor the decrease of macular volume attentively? Mention why medically this is clinically impactful.

But then state that on the evidence you observed in your study more data needs to be collected to make a statistically robust inference of the true size of the effect. Is this a very good hypothesis generating and worthy of future, detailed, studies.